

**Formation of Chlorinated Organic Compounds from Cl Atom-Initiated**
**Reactions of Aromatics and Their Detection in Suburban Shanghai**
**Chuang Li[1], Lei Yao[1,2]\*, Yuwei Wang[1], Mingliang Fang[1], Xiaojia Chen[1], Lihong Wang[1],**
**Yueyang Li[1], Gan Yang[1], Lin Wang [1,2,3,4,5]\***
[1] Shanghai Key Laboratory of Atmospheric Particle Pollution and Prevention (LAP[3]),
Department of Environmental Science and Engineering, Jiangwan Campus, Fudan
University, Shanghai 200438, China
[2] Shanghai Institute of Pollution Control and Ecological Security, Shanghai 200092, China
[3] IRDR International Center of Excellence on Risk Interconnectivity and Governance on
Weather/Climate Extremes Impact and Public Health, Fudan University, Shanghai 200438,
China
[4] National Observations and Research Station for Wetland Ecosystems of the Yangtze Estuary,
Shanghai, 200433, China
[5] Collaborative Innovation Center of Climate Change, Nanjing, 210023, China
*Corresponding Author: L.Y., email, lei_yao@fudan.edu.cn; phone, +86-21-31243568*
*L.W., email, lin_wang@fudan.edu.cn; phone, +86-21-31243568*
**Abstract.** Chlorine (Cl) atoms generated from the photolysis of atmospheric reactive
chlorine species can rapidly react with various volatile organic compounds (VOCs), forming
chlorine- and non-chlorine-containing low-volatile oxygenated organic molecules. Yet, the
formation mechanisms of chlorine-containing oxygenated organic molecules (Cl-OOMs) from
reactions of Cl atoms with aromatics in the presence and absence of NO$_x$ are not fully
understood. Here, we investigated Cl-OOMs formation from Cl-initiated reactions of three
typical aromatics (i.e., toluene, m-xylene, and 1,2,4-trimethylbenzene (1,2,4-TMB)) in the
laboratory and searched for ambient gaseous Cl-OOMs in suburban Shanghai. From our
laboratory experiments, 19 Cl-containing peroxyl radicals and a series of Cl-OOMs originating
from the Cl-addition-initiated reaction were detected, which provides direct evidence that the
Cl-addition-initiated reaction is a non-negligible pathway. In addition, a total of 51 gaseous Cl-
OOMs were identified during the winter in suburban Shanghai, 38 of which were also observed
in laboratory experiments, hinting that Cl-initiated oxidation of aromatics could serve as a
source of Cl-OOMs in an anthropogenically influenced atmosphere. Toxicity evaluation of



33 these Cl-OOMs shows potential adverse health effects. These findings demonstrate that Cl-

34 OOMs can be efficiently formed via the Cl-addition pathway in the reactions between aromatics

35 and Cl atoms and some of these Cl-OOMs could be toxic.

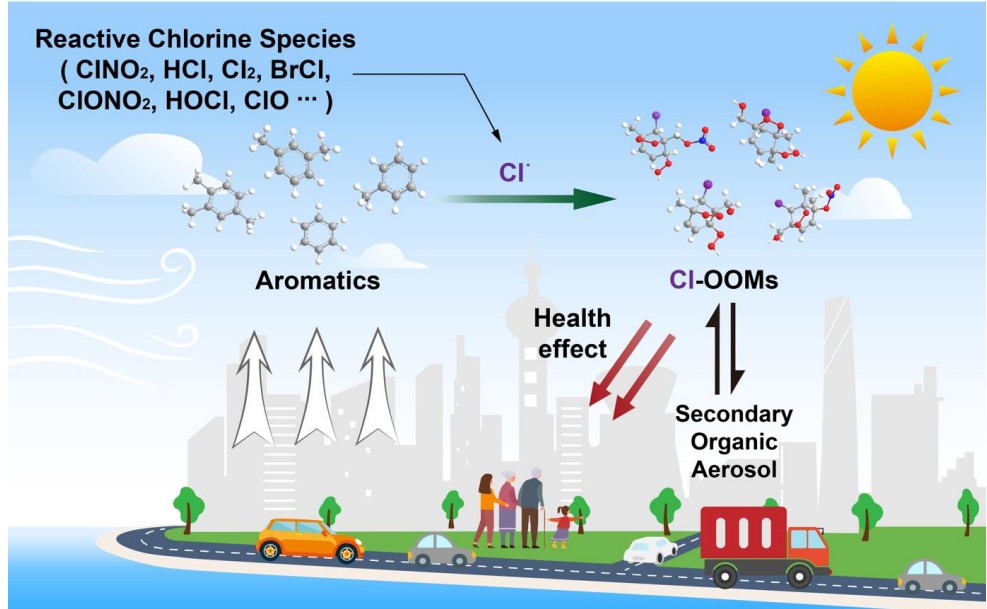




## 1. INTRODUCTION


Atmospheric chlorine atoms (Cl), together with hydroxyl radicals (OH), ozone ($O_3$), and nitrate
radicals ($NO_3$), play vital roles in transforming volatile organic compounds (VOCs), leading to
the formation of oxygenated organic molecules (OOMs) and secondary organic aerosol
(SOA)(Priestley et al., 2018; Shang et al., 2021; Tham et al., 2016; Thornton et al., 2010). The
involvement of Cl atoms in atmospheric chemical processes was conventionally thought to be
confined to the oceanic boundary layer (Keene et al., 1999; Knipping et al., 2000). Estimated
Cl atom concentrations in the coastal region ranged from $10^2$ to $10^5$ molecules $cm^{-3}$ (Thornton
et al., 2010; Wingenter et al., 2005). Recently, a number of reactive chlorine species, such as
nitryl chloride ($ClNO_2$), chlorine nitrate ($ClONO_2$), hypochlorous (HOCl), chlorine ($Cl_2$),
bromine chloride (BrCl), and hydrochloric acid (HCl), were found to lead to high
concentrations of Cl atoms in the urban and suburban atmospheres (Breton et al., 2018; Peng
et al., 2020; Priestley et al., 2018). During the daytime, peak Cl atom concentrations can reach
$10^6$ molecules $cm^{-3}$ (Wang et al., 2023). which is still less than the global average concentrations
of OH radicals (Breton et al., 2018; Liu et al., 2017). Nonetheless, the reaction rate coefficients
of VOCs with Cl atoms are generally 1-2 orders of magnitude larger than those of OH radicals,
which can partially compensate for the lower concentration of Cl atoms when determining the
relative importance of different reactive loss pathways of VOCs (Chen et al., 2023; Riva et al.,
2015; Wang et al., 2005).
Apart from H-abstraction, Cl atoms can be added to VOCs forming chlorine-containing
oxygenated organic molecules (Cl-OOMs) in reactions of VOCs with Cl atoms. For small
alkenes (e.g., isoprene), the reaction mechanism is dominated by Cl-addition to the double bond,
with some allylic hydrogen abstraction (approximately 15% for isoprene at 1 atm) (Finlayson-
Pitts et al., 1999; Orlando et al., 2003; Ragains and Finlayson-Pitts, 1997). For larger biogenic
VOCs (e.g., β-pinene), which contain a greater number of abstractable hydrogen atoms, H-
abstraction becomes more significant; for instance, H-abstraction is thought to account for half
of the overall initiation reactions (Finlayson-Pitts et al., 1999). Overall, both Cl-addition and
H-abstraction pathways coexist for biogenic VOCs, with the Cl-addition pathway possibly



being the more dominant pathway, such as isoprene (Wang et al., 2022) and d-limonene (Wang
et al., 2019).

Given the ubiquitous existence of aromatics in the urban air and the recent detection of

reactive chlorine species that are precursors of Cl atoms in the same atmosphere, the reaction
mechanisms of aromatics and Cl atoms should be of concern. Although the reaction rate
coefficient of benzene with Cl atoms under normal atmospheric conditions is relatively slow
$(1.3 \times 10^{-15}$ cm$^3$ molecule$^{-1}$ s$^{-1}$) and this reaction is insignificant in the ambient atmosphere, its
reaction mechanism provides valuable insights into reaction pathways of Cl atoms and
aromatics (Shi and Bernhard, 1997). Sokolov *et al.* proposed that $C_6H_6Cl$ radicals, formed
through a Cl-addition pathway in reactions of Cl atoms and benzene, can either decompose
back to benzene or further react in a non-aromatizing manner (Sokolov et al., 1998). In contrast,
toluene, xylene, and trimethylbenzene react with Cl atoms at faster rates, urging a better
understanding of their reaction mechanisms. Using the density functional theory and the
conventional transition state theory, Huang *et al.* investigated the Cl-toluene reaction (Huang
et al., 2012). Their findings indicate that the reaction rate coefficient for the H-abstraction
pathway $(5.58 \times 10^{-11}$ cm$^3$ molecule$^{-1}$ s$^{-1}$) is substantially higher than that for the Cl- addition
pathway $(0.91 \times 10^{-11}$ cm$^3$ molecule$^{-1}$ s$^{-1}$), highlighting the significance of the H-abstraction
pathway that accounts for approximately 86% of Cl-initiated reactions. Studies employing gas
chromatography-mass spectrometry (GC-MS) to examine products of Cl-initiated reactions of
toluene and iodide-based chemical ionization mass spectrometry (I-CIMS) to analyze products
of Cl-initiated reactions of m-xylene, have both predominantly detected chlorine-free
oxygenated organic compounds, which were supposed as main products (Cai et al., 2008; Wang
et al., 2005). These results have guided atmospheric models to integrate the H-abstraction as
the primary reaction pathway in their analytical frameworks (Ma et al., 2023; Peng et al., 2022).
However, studies focusing on the Cl-addition pathway and its related products are sparse, and
the significance of the Cl-addition pathway in the atmospheric reactions of Cl atoms and
aromatics remains elusive. Over the past decade, thanks to the development of nitrate-based
chemical-ionization atmospheric-pressure-interface long-time-of-flight mass spectrometers





(nitrate-CI-APi-LToF), there have been significant advancements in the detection of highly
oxygenated organic molecules and radicals, which is crucial for elucidating the mechanisms of
Cl-initiated reactions with VOCs (Bianchi et al., 2019; Ehn et al., 2010). The formation of
highly oxygenated organic products, including Cl-containing ones from the reactions of
biogenic VOCs with Cl atoms, were gradually revealed, whereas whether or not significant
amounts of Cl-containing highly oxygenated organics and radicals can be formed from Cl atoms
and aromatics remains unclear (Wang et al., 2020).
OOMs can lead to potential air quality, climate, and health effects. Due to their low volatility,
OOMs have been identified as dominant precursors for the growth of newly formed particles
and formation of SOAs, which are known for their negative effect on air quality and climate
impacts (Ehn et al., 2014; Kulmala et al., 2013). Compared to non-chlorine-containing OOMs
(non-Cl-OOMs), Cl-OOMs formed through the introduction of chlorine substituents make
organic compounds more lipophilic, facilitating their interactions with hydrophobic sites and
promoting enzymatic biotransformation in general, which can lead to adverse health effect in
turn (Henschler, 1994). Therefore, field and laboratory studies for characteristics and sources
of Cl-OOMs from the reactions of Cl-aromatics and their risk assessment upon human
atmospheric exposure should be carried out.
In this study, we investigated non-Cl-OOMs and Cl-OOMs formation mechanisms from Cl-
initiated reactions of toluene, m-xylene, and 1,2,4-trimethylbenzene (1,2,4-TMB) in the
presence and absence of $NO_x$ in a laboratory flow reactor, using a nitrate-CI-APi-LToF
(Aerodyne Research, Inc. USA, and Tofwerk AG, Switzerland) and a Vocus proton-transfer-
reaction long-time-of-flight mass spectrometer (Vocus-PTR-LToF) (Tofwerk AG,
Switzerland). In addition, the nitrate CI-APi-LToF was also deployed in a field campaign in
suburban Shanghai during winter to search for ambient gaseous Cl-OOMs. The toxicity of
selected Cl-OOMs, which were simultaneously detected both in laboratory experiments and
ambient observations, was evaluated by computational toxicity.

## 2. MATERIALS AND METHODS

### 2.1. Experimental Set-up in the Laboratory.



A general scheme of the experimental setup is shown in Figure S1. Simulation experiments
were conducted in a 6 L quartz flow tube reactor with a total flow rate of 10 L min$^{-1}$, resulting
in a residence time of ~36 seconds. This flow tube is covered by aluminum composite panels
to avoid room light. Zero air with relative humidity (RH) less than 1% generated from a Zero
Air Generator (AADCO Instruments, Inc. USA) was used as carrier gas. The reaction
temperature was maintained at around 20°C.
Gaseous aromatics were prepared from their standards (toluene, $\geq$ 99.0%, Aladdin; m-xylene,
$\geq$ 99.0%, Aladdin; 1,2,4-TMB, $\geq$ 99.5%, Aladdin) together with high-purity nitrogen gases. Cl
atoms were produced by photolysis of chlorine (Cl$_2$, Shanghai Wetry Standard Reference Gas
Analytical Technology Co., LTD) using 350 nm UV lights. In experiments with NO$_x$, NO (Air
Liquid Co., LTD) was added into the flow tube to produce and sustain NO$_x$ mixing ratios that
were sufficiently high to be a competitive sink for RO$_2$ radicals. RH is controlled by changing
zero air flowrates through the water bubbler. Before each experiment, the wall of the flow tube
was cleaned with a water/alcohol solution and then purged with zero air for over 1 hour.
The concentration of Cl atoms was controlled by adjusting the flow rate of Cl$_2$. The mean
concentrations of Cl atoms were determined according to Eq.(1) using reaction rate coefficients
$k$ of 6.2×10$^{-11}$ cm$^3$ molecule$^{-1}$ s$^{-1}$, 1.35×10$^{-10}$ cm$^3$ molecule$^{-1}$ s$^{-1}$, and 2.42×10$^{-10}$ cm$^3$ molecule$^{-1}$
s$^{-1}$ for reactions between Cl atoms and toluene, m-xylene, and 1,2,4-TMB, respectively(Wang
et al., 2005), as follows:
$$[Cl] = -1/kt \times \ln([Aromatics]_t/[Aromatics]_0) \qquad \text{Eq.} (1)$$
where $[Aromatics]_0$ and $[Aromatics]_t$ are the initial concentration and the concentration
after a reaction time $t$ of aromatic precursors, respectively. $[Cl]$ is the estimated concentration
of Cl atoms in the flow tube. In our flow tube experiments, the extent of oxidation is quantified
using the parameter of Cl exposure, defined as $[Cl]$ multiplied by the reaction time $t$. Cl
exposures in our experiments were in the range of (1.2-2.0) ×10$^9$ molecules cm$^{-3}$ s, equivalent
to atmospheric oxidation times of roughly 0.6-1.3 hours for aromatics at a daytime Cl atom
concentration of 5×10$^5$ molecules cm$^{-3}$ (Chang et al., 2004; Tham et al., 2016; Wang et al.,

2023).



A Vocus-PTR-LToF and a nitrate-CI-APi-LToF (more details refer to Text S1 & Figure S2
in Supplemental Information) were simultaneously deployed to detect aromatic precursors and
gaseous OOM products, respectively. Their working principles were described in details
elsewhere (Eisele and Tanner, 1993; Krechmer et al., 2018). Signals of aromatic precursors and
reaction products measured from the zero air were treated as their background. The resolving
power of the nitrate CI-APi-LToF was up to around 8000 for ions with m/z larger than 200 Th.
The ions of $NO_3^-$, $HNO_3 \cdot NO_3^-$, and $C_6H_5NO_3 \cdot NO_3^-$ were selected for mass calibration, and the
calibration error is less than 1ppm. When identifying the OOM signal peaks, the error is limited
below 4 ppm.
OOM concentrations are estimated by Eq. (2)(Kürten et al., 2016),

$$[OOMs] = C \times \frac{OOM \cdot NO_3^-}{NO_3^- + HNO_3 \cdot NO_3^- + (HNO_3)_2 \cdot NO_3^-} \times T \qquad \text{Eq. (2)}$$


where $OOM \cdot NO_3^-$, $NO_3^-$, $HNO_3 \cdot NO_3^-$, and $(HNO_3)_2 \cdot NO_3^-$ represent signals of corresponding
ions in units of counts per second (cps). OOMs with an oxygen content of equal to or more than
6 (i.e., highly oxygenated organic molecules, HOMs) are assumed to cluster with $NO_3^-$ at the
same rate coefficient as that of sulfuric acid ($H_2SO_4$), i.e., both at collision-limited rates
(Bianchi et al., 2019; Ehn et al., 2010). Therefore, the calibration factor $C$ for sulfuric acid is
adopted as that of OOMs (Kürten et al., 2011, 2012). It should be noted we also used the same
calibration factor C for quantification of OOMs with an oxygen number of less than 6, which
may lead to relatively high uncertainties (Alage et al., 2024). A mass-dependent transmission
correction factor $T$ of our instrument is also taken into account in this study (Heinritzi et al.,
2016). The mass-dependent transmission correction factor is instrument-specific and
determined by depleting the primary ion with a series of perfluorinated acids and comparing
the primary ion signal depletion with the product signal increase (which would match for
equivalent transmission efficiency) (Lu et al., 2020).
In addition, a $NO_x$ monitor (Thermo, 49i) was utilized to measure $NO_x$ concentrations in
laboratory experiments. A nano-SMPS (Scanning Mobility Particle Sizer with a nano
Differential Mobility Analyzer, TSI, USA) together with a PSM (Particle Size Magnifier,





Airmodus, Finland) were used to detect particles in the range of sub-3 nm to 60 nm, indicating
the absence of newly formed particles during all experiments.

Table 1 summarizes experimental conditions including mixing ratios of aromatic precursors

(i.e., toluene, m-xylene, and 1,2,4-TMB), $NO_x$, and RH.



**Table 1.** Summary of experimental conditions in the laboratory experiments.

| Exp. | Precursor | Initial precursor concentration (ppb) | Initial NO[a] (ppb) | Estimated Cl exposure (×10⁹ molecule cm⁻³ s) | RH[b] (%) | Non-Cl-OOMs molar yield (%) | Cl-OOMs molar yield (%) | Ratio (Cl-OOMs/ Total OOMs, %) | Ratio (Dimer/ Monomer, %)[c] |
|------|-----------|------------------|----------|--------------------------|---------|------------------|----------------|------------------|------------------|
| 1 | Toluene | 80 | 0 | 2.0 | <1 | 5.1 | 2.4 | 32 | 3.5 |
| 2 | Toluene | 80 | 0 | 1.2 | 60 | 0.7 | 0.3 | 31 | 2.0 |
| 3 | Toluene | 84 | 45 | 2.7 | <1 | 6.8 | 2.6 | 28 | 1.7 |
| 4 | Toluene | 80 | 40 | 1.6 | 60 | 1.8 | 0.8 | 29 | 1.0 |
| 5 | m-Xylene | 87 | 0 | 2.0 | <1 | 0.6 | 0.5 | 43 | 3.2 |
| 6 | m-Xylene | 87 | 0 | 1.6 | 68 | 0.3 | 0.2 | 44 | 2.6 |
| 7 | m-Xylene | 90 | 45 | 1.7 | <1 | 1.4 | 0.6 | 31 | 0.9 |
| 8 | m-Xylene | 87 | 50 | 2.3 | 35 | 1.1 | 0.5 | 32 | 1.2 |
| 9 | 1, 2, 4-TMB | 98 | 0 | 1.4 | <1 | 0.5 | 0.3 | 34 | 2.1 |
| 10 | 1, 2, 4-TMB | 103 | 0 | 1.4 | 30 | 0.3 | 0.2 | 35 | 1.4 |
| 11 | 1, 2, 4-TMB | 93 | 40 | 2.2 | <1 | 1.5 | 0.7 | 31 | 0.8 |
| 12 | 1, 2, 4-TMB | 109 | 55 | 2.2 | 30 | 1.0 | 0.6 | 37 | 0.6 |

[a] In the presence of NO experiments, there is OH chemistry was involved in Cl-aromatics reactions and its influence can lead to relatively high uncertainty in the molar
yields reported in the table for pathways influenced by OH chemistry. Approximately 90% of NO was converted into NO₂ after the UV light was turned on.
[b] Relative humidity.
[c] The molar yield ratios of total OOM dimers to monomers. Monomers are defined as molecules with a carbon number equal to the carbon number of aromatic precursor
(nC), and dimers are defined as molecules with carbon numbers ranging from 2nC -1 to 2nC +1.



**2.2. Field measurements.**
A field campaign was conducted from December 14[th], 2022, to February 2[nd], 2023, at the
Dianshan Lake (DSL) Air Quality Monitoring Supersite in suburban Shanghai, China
(31.10°N,120.98°E). This monitoring site is frequently impacted by regional transport and
experiences episodes of anthropogenic pollution. A detailed description of this site can be found
elsewhere (Wu et al., 2023; Yang et al., 2022, 2023). A Vocus-PTR-LToF and a nitrate-CI-
APi-LToF were both deployed in this field campaign to detect aromatics and Cl-OOMs,
respectively. The detailed description of nitrate-CI-APi-LToF and Vocus-PTR-LToF, and their
calibration in the field measurement are shown in the Supplemental Information (Text S2).
**2.3. Heath effect estimation.**
A number of Cl-OOMs were assessed using the Estimation Program Interface Suite (EPI, V.
4.11) and Toxicity Estimation Software Tool (T.E.S.T, V. 5.1.2) software provided by the
United States Environmental Protection Agency (EPA), to estimate their persistence,
bioaccumulation, and toxicity through calculated half-life for reactions with OH,
bioconcentration factors (BCF), oral rat $pLD_{50}$ (-log10(pred), mol/kg), developmental toxicity,
and mutagenicity. The models utilized the SMILES (Simplified Molecular Input Line Entry
System) notation of the target compounds as input for the prediction.
## 3.   RESULTS AND DISCUSSION
**3.1. OOM molar yields.**
The molar yields of OOMs are determined as OOMs formed ($\Delta M$, molar cm$^{-3}$) divided by
precursor reacted ($\Delta Ar$, molar cm$^{-3}$):

$$Molar\ yield = \frac{\Delta M}{\Delta Ar} \qquad\qquad Eq.(3)$$


Table 1 summarizes non-Cl-OOMs' and Cl-OOMs' molar yields and OOMs' dimer-to-monomer
ratios from our laboratory experiments. The molar yields of non-Cl-OOMs and Cl-OOMs from
reactions of three aromatic precursors with Cl atoms in the absence of $NO_x$ are within the ranges
of 0.3-5.1% and 0.2-2.4%, respectively. These values are comparable with previously reported:
HOM molar yields (0.8-4.0%) detected by nitrate-CI-APi-LToF for the reactions between α-





pinene and Cl atoms (Wang et al., 2020) and non-Cl-OOMs molar yields (4.4-8.8%) detected
by $H_3O^+$-Chemical Ionization Mass Spectrometry ($H_3O^+$-CIMS) for the reactions between m-
xylene and Cl atoms (Bhattacharyya et al., 2023). The low molar yields observed in both this
study and previous studies may be attributed to the preference of different detection techniques.
Nevertheless, the observed ratio of Cl-OOMs to non-Cl-OOMs, ranging from 29% to 44%
(Table 1), further indicates the non-negligibility of the Cl-OOMs products among the total
OOMs products. In the presence of 40-55 ppb $NO_x$, our flow tube experiments show that the
molar yields of non-Cl-OOMs and Cl-OOMs from three aromatics are within the ranges of 1.0-
6.8% and 0.5-2.6%, respectively. The addition of $NO_x$ can slightly increase the molar yields of
both non-Cl-OOMs and Cl-OOMs.

Typically, the fate of peroxy radicals in flow tube experiments is largely influenced by their

reactions with other $RO_2$, $HO_2$, or NO species, which are contingent upon the specific
experimental conditions (Bianchi et al., 2019; DeMore et al., 1997). In experiments without
$NO_x$, while the termination of $RO_2$ was primarily anticipated to be governed by $RO_2$-$RO_2$ and
$HO_2$-$RO_2$ reactions. However, after the addition of $NO_x$, the reaction between $RO_2$ and NO
tended to predominate, leading to a reduction in the dimer-to-monomer ratio by 50.5% (Exp. 3
and Exp. 1 in Table 1).

In contrast to OH reaction, the presence of Cl, ClO, and $Cl_2$ in chlorine-involved reaction

may introduce additional reaction pathways related to Cl-OOM formation. However, these
pathways make only minor contributions to the formation of Cl-OOMs under our experimental
conditions (see Text S4 in Supplemental Information for detailed analysis).

Although both the non-Cl-OOMs and Cl-OOMs molar yields increased in the presence of

$NO_x$, the ratio of Cl-OOMs to the total OOMs decreased when $NO_x$ was added. This
phenomenon can be ascribed to the critical involvement of OH chemistry, stemming from the
NO+$HO_2$ reaction. The additional OH radicals can contribute to the formation of non-Cl-OOMs,
as supported by recent experimental and modeling studies on the reaction dynamics between
Cl atoms and isoprene (Wang et al., 2022). For our experiments, as $NO_x$ was added into the
flow tube, the concurrent presence of Cl atoms, OH radicals, and aromatics led to a notable





increase in the total OOM molar yields, ranging from 25 to 187%, while reducing the ratios of
Cl-OOMs to the total OOMs by approximately 9-28% (Table 1). It should be noted that molar
yields reported in Table 1 with high uncertainty due to OH chemistry in the presence of NOx.
Besides, high RH leads to low molar yields, which may be attributed to the depressed detection
efficiency of OOMs and the elevated vapor wall loss under humid conditions (Huang et al.,
2018). The product distribution remains unchanged under high humidity conditions (see Figure
1B and Figure S3), indicating that the presence of water does not significantly influence the
reaction between Cl atoms and aromatics.
**3.2. Characteristics of OOM products and peroxyl radicals.**
Mass defect plots of stabilized products from reactions between toluene (Exp.1 in Table1),
m-xylene (Exp.5 in Table1), and 1,2,4-TMB (Exp.9 in Table1) and Cl atoms in the absence of
NO$_x$ are shown in Figure 1. These products display similar distribution patterns, consisting of
monomers (with carbon numbers equal to carbon numbers of the precursor, nC) and dimer
products (with carbon numbers ranging from 2nC -1 to 2nC +1). The ratio of dimer products to
monomer products are 3.5%, 3.2%, and 2.1% for reactions of toluene, m-xylene, and 1,2,4-
TMB with Cl atoms, respectively. Meanwhile, the products also can be classified into two
groups: non-Cl-OOMs in blue and Cl-OOMs in orange in Figure 1. In general, the total
concentration of Cl-OOMs is lower than that of non-Cl-OOMs. Specifically, the concentrations
of Cl-OOMs account for 47%, 91%, and 52% of the non-Cl-OOMs in the toluene, p-xylene,
and 1,2,4-trimethylbenzene experiments, respectively. Both non-Cl-OOMs and Cl-OOMs
products can be categorized into several bands, as indicated by the dashed lines in Figure 1,
each of which comprises compounds with varying numbers of oxygen atoms.



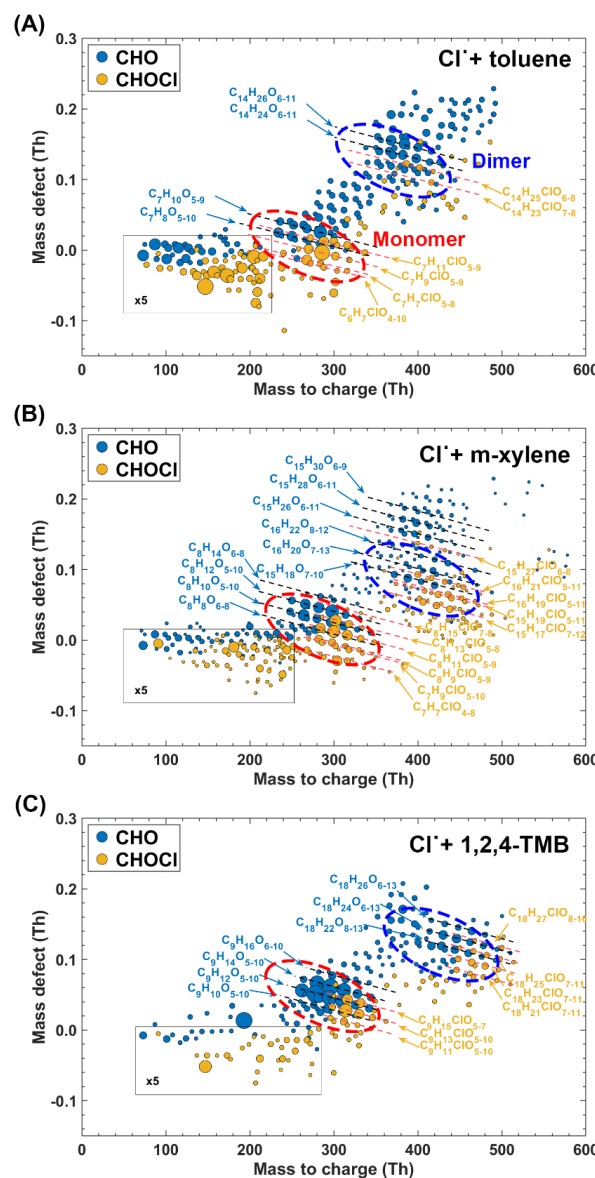

**Figure 1.** Mass defect plots of OOM products detected by a nitrate-CI-APi-LTOF from Cl-initiated reactions of (A) toluene, (B) m-xylene, and (C) 1,2,4-trimethylbenzene, respectively, in the absence of $NO_x$. The detected products are marked by their exact mass (with $NO_3^-$ reagent ions) and mass defect (exact mass subtracted by its unit mass). The lines annotate the general chemical formulae. Chlorine-containing and non-chlorine-containing formulae are shown in different colors. The size of the circle corresponds to the concentration of products.



The mass spectrum of the detected OOMs monomer and dimer products from the reaction

between m-xylene and Cl atoms with and without $NO_x$ (Exp. 5&7 in Table 1) are shown in

Figure 2. Without $NO_x$, dominant monomer products for non-Cl-OOMs included $C_8H_{10}O_6$ and

$C_8H_{12}O_{6-8}$, whereas for Cl-OOMs, $C_8H_{11}ClO_{6-8}$ and $C_8H_{13}ClO_{6-7}$ dominated. The most

abundant dimer compounds for non-Cl-OOMs were $C_{16}H_{20}O_9$, $C_{16}H_{22}O_{10}$, and $C_{15}H_{18}O_8$,

whereas for Cl-OOMs, $C_{16}H_{19}ClO_{8,10}$, $C_{16}H_{21}ClO_9$, and $C_{15}H_{17}ClO_8$ prevailed. Dimer products

containing two Cl atoms were also observed, exemplified by $C_{16}H_{18,20}Cl_2O_{10}$. Under $NO_x$-

present conditions, the main non-Cl-OOMs included $C_8H_{11}NO_{6-10}$ and $C_8H_{12}N_2O_{10}$, whereas

Cl-OOMs were represented by $C_8H_{10}ClNO_{7-8}$ and $C_8H_{11}ClN_2O_{9-10}$.


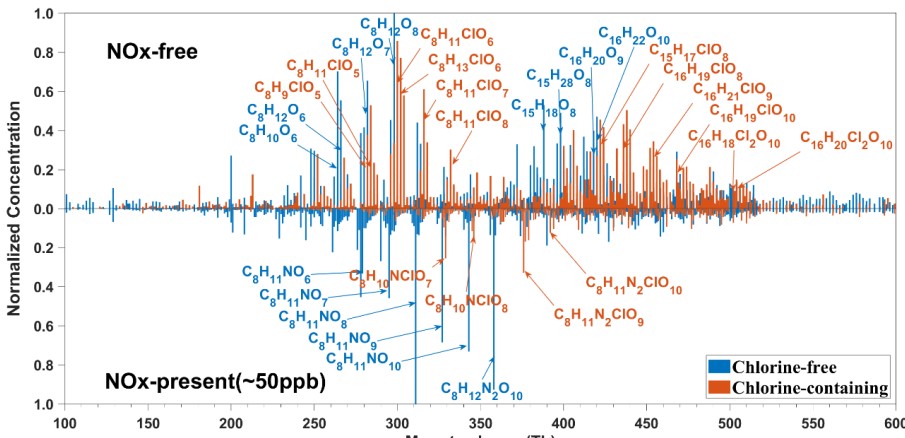

**Figure 2.** Mass spectra of OOM products detected by a nitrate-CI-APi-LToF from the reaction of m-xylene and
Cl atoms under $NO_x$-free and $NO_x$-present conditions. The y-axes in both figures are standardized by setting their
maximum concentrations to 1.

Moreover, 26 peroxyl radicals in total were observed in our flow tube experiments with three

precursors, as listed in Table S1 and illustrated in Figures 3 & S4. It is crucial to note, however,

that not every intermediate radical formed could be conclusively identified. This is primarily

due to the inherent instability and the extremely brief life span of peroxyl radicals, which pose

significant challenges for their detection.

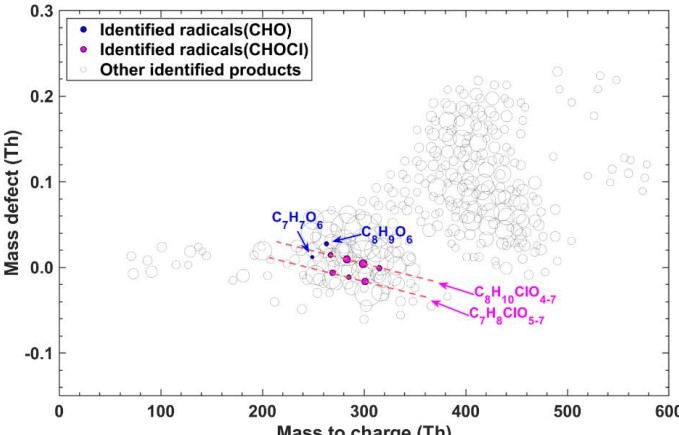

**Figure 3.** Mass defect plot of peroxyl radicals detected by a nitrate-CI-APi-LToF from Cl-initiated reactions of m-xylene without $NO_x$. The detected products are marked by their exact mass (with $NO_3^-$ reagent ions) and mass defect (exact mass subtracted by its unit mass). The lines annotate the general chemical formulas. Chlorine-containing and non-chlorine-containing formulas are shown in different colors. The size of the circle is proportional to the concentrations of peroxyl radicals.

Take the reaction between Cl atoms and m-xylene in the absence of $NO_x$ for example, we
detected peroxyl radicals including $C_8H_{10}ClO_{4-7}$, $C_7H_8ClO_{5-7}$, $C_8H_9O_6$, and $C_7H_7O_6$ (Figure 3).
High-resolution peak fittings of partial peroxy radicals from the raw mass spectrum obtained
by nitrate-CI-APi-LToF are shown in Figure S5. Notably, the dominant species among these
radicals was $C_8H_{10}ClO_6$, constituting 27.4% of the total signal of identified peroxyl radicals in
the reaction of Cl atoms and m-xylene and nearly three times of the $C_8H_9O_6$ radical signal. Also,
similar ratio rules of chlorine-containing radicals (analogy of $C_8H_{10}ClO_6$) to non-chlorine-
containing radicals (analogy of $C_8H_9O_6$) were observed for toluene (~4.2 times) and 1,2,4-TMB
(~3.8 times). Although $C_7H_8ClO_{5-7}$ radicals were also discernible, their signal intensity was
merely 22.3% of the total signal of identified peroxyl radicals in the reaction of Cl atoms and
m-xylene. Non-Cl-containing radicals were also detected, albeit with substantially lower signal
values, accounting for only 15.5% of the total identified radical signals.
**3.3. Formation mechanisms of Cl-OOMs**
**3.3.1. Cl-addition pathway**





Given the similar product patterns for the three precursors (toluene, m-xylene, and 1,2,4-
TMB), it is reasonable to infer that the reaction mechanisms between different aromatics and
Cl atoms are analogous. A generalized mechanism is thus proposed, elucidating the Cl-initiated
reactions of m-xylene in the absence of NOx, as depicted in Scheme 1. This scheme serves as
a representative example highlighting the potential pathways involved in the Cl-initiated
reactions of aromatics.

**Scheme 1.** Proposed reaction mechanisms of m-xylene with Cl atoms leading to the formation of OOMs. Blue
and black formulae denote radicals, and stable products, respectively. Radicals and stable products detected by
nitrate-CI-APi-LToF are marked with black boxes.
Although the theoretical studies by Huang *et al.* (2012) show that the Cl-addition pathway
accounts for only 14% of Cl-initiated reactions of toluene (298K), which is significantly lower
than the 86% attributed to the H-abstraction pathway, the proportion of Cl-OOMs products
from Cl-addition reaction should not be overlooked, as evidenced by the observation of peroxyl
radicals and product distribution characteristics in this study (Huang et al., 2012). The initial
reaction of m-xylene ($C_8H_{10}$) with Cl atoms can occur through two pathways: the addition
pathway, Cl-addition leading to the formation of a $C_8H_{10}Cl$ radical ($C_8H_{10}Cl\cdot$) or the H-
abstraction pathway, forming a $C_8H_9$ radical ($C_8H_9\cdot$) (Scheme 1). Then, both $C_8H_9\cdot$ and
$C_8H_{10}Cl\cdot$ can in turn undergo autoxidation via the formal addition of $O_2$ to produce peroxy
radicals of $C_8H_9O_6\cdot$ or $C_8H_{10}ClO_6\cdot$. The peroxyl radical $C_8H_{10}ClO_6\cdot$ was identified as the
predominant species in terms of signal (Table S1). The signal intensity ratio of $C_8H_{10}ClO_6\cdot$ to
$C_8H_9O_6\cdot$ was around 3 in our reaction, which suggests that Cl-addition pathway could be at
least a non-negligible pathway in the initial reaction steps of Cl atoms and m-xylene, compared



with H-abstraction pathway. Furthermore, the signal intensity ratios of Cl-addition pathway
radials ($C_7H_8ClO_6\cdot$ from toluene and $C_9H_{12}ClO_4\cdot$ from 1,2,4-TMB) and H-abstraction pathway
radicals ($C_7H_7O_6\cdot$ from toluene and $C_9H_{11}O_4\cdot$ from 1,2,4-TMB) were 4.2 and 5.6 (Table S1),
respectively. Although these Cl-OOMs-to-non-Cl-OOMs signal ratios may not accurately
represent their relative concentrations due to sensitivity differences of these radicals towards
the reagent ions ($(HNO_3)_{0-1}\cdot NO_3^-$), it is still noteworthy that these Cl-$RO_2$ overlooked in
previous studies were directly observed in such a reaction system, thereby suggesting that Cl-
addition pathway is indeed present in the initial reaction steps of reactions between Cl atoms
and aromatics.

In the reaction of Cl atoms and m-xylene, a certain fraction of the peroxyl radicals

($C_8H_{10}ClO_6\cdot$) might also be derived from the secondary Cl-addition reaction between Cl atoms
and a first-generation stabilized product $C_8H_{10}O_6$ (Scheme 1). Indeed, it is challenging to
evaluate the exact contribution from secondary Cl-addition reaction to the formation of
$C_8H_{10}ClO_6\cdot$. However, it may be indirectly assessed via the potential secondary reactions
between more dominant first-generation stabilized products ($C_8H_{12}O_x$) and Cl (Table S2), since
$C_8H_{10}O_6$ and $C_8H_{12}O_x$ likely react with Cl at similar rates. If secondary Cl-addition reactions
were significant in the reaction system of Cl atoms and m-xylene, $C_8H_{12}O_x$ should undergo
secondary Cl-addition reactions to generate $C_8H_{12}ClO_x$ radicals. Yet, $C_8H_{12}ClO_x$ radicals were
not detectable, which hints that secondary Cl-addition reactions could only play a minor role in
our experiments. Therefore, it appears that the secondary Cl-addition reactions between
stabilized products and Cl atoms are less significant compared with Cl-addition in the initial
reaction steps.

Recently, Jahn et al. (2024) investigated the formation of secondary organic aerosols from

the reaction of ethylbenzene and Cl atoms, and attributed the observed Cl-addition products to
reactions involving non-aromatic C=C bonds (Jahn et al., 2024). They claimed that OH radical
existed in their experiments, due to the presence of NOx. As a result, approximately 40% of
ethylbenzene reacted with Cl atoms, 30% with OH, and 30% remained unreacted. Thus, a
secondary Cl addition to $C_8H_{11}NO_6$ for reaction of ethylbenzene and Cl atoms, forming



$C_8H_{12}ClNO_8$, is possible (Scheme S1). However, OH radical was expected to be minor at least
in our NOx-free experiments. Thus $C_8H_{11}NO_6$, as an OH addition product (Scheme S2), was
insignificant in our NOx-free reaction of m-xylene and Cl atoms (refer to Figure 2), leading to
a minor role of a secondary Cl addition to form $C_8H_{12}ClNO_8$. In our experiments with NOx, the
primary Cl-contain products, i.e., $C_8H_{10}ClNO_7$ (Scheme S3), contain two less hydrogen atoms
than $C_8H_{12}ClNO_8$, likely explained by the existence of one more double bond in the structure
of $C_8H_{10}ClNO_7$. The presence of this additional double bond excludes the possibility of a
sequential OH addition and Cl addition.
**3.3.2. Autoxidation and subsequent reactions of Cl-RO$_2$**
Scheme 2 shows a proposed reaction mechanism for the autoxidation of the main Cl-RO$_2$
radical ($C_8H_{10}ClO_4\cdot$) generated from Cl atoms and m-xylene. Autooxidation of $C_8H_{10}ClO_4\cdot$
leads to $C_8H_{10}ClO_6\cdot$. There are two distinct isomeric forms of $C_8H_{10}ClO_6\cdot$, denoted as
$C_8H_{10}ClO_6\cdot$(I) and $C_8H_{10}ClO_6\cdot$(II), which is similar to results for aromatics + OH by Molteni
et al (Molteni et al., 2018). Notably, the formation of $C_8H_{10}ClO_6\cdot$(II) requires a second step of
endo-cyclization, which is not competitive on account of its slow reaction rate, as inferred from
several previous studies using both experimental and theoretical approaches of OH-initiated
oxidation of aromatics (Wang et al., 2017; Xu et al., 2020). Therefore, the abundance of
$C_8H_{10}ClO_6\cdot$(I) would likely be much higher than $C_8H_{10}ClO_6\cdot$(II), and our following discussion
primarily focuses on the subsequent reaction processes involving $C_8H_{10}ClO_6\cdot$(I). Briefly,
$C_8H_{10}ClO_6\cdot$ reacts with RO$_2\cdot$ to generate $C_8H_{9,11}ClO_5$ and $C_8H_{10}ClO_5\cdot$ (Reaction pathway *R2b*)
and with an HO$_2$ radical to produce $C_8H_{11}ClO_6$ (Reaction pathway *R2c*). Meanwhile,
$C_8H_{10}ClO_6\cdot$ has no more H atoms for another H-shift at an appreciable rate based on our current
understanding.(Wang et al., 2024) In the presence of NO$_x$, $C_8H_{10}ClO_6\cdot$ is terminated (Reaction
pathway *R2a*), leading to the formation of $C_8H_{10}ClNO_7$ products.



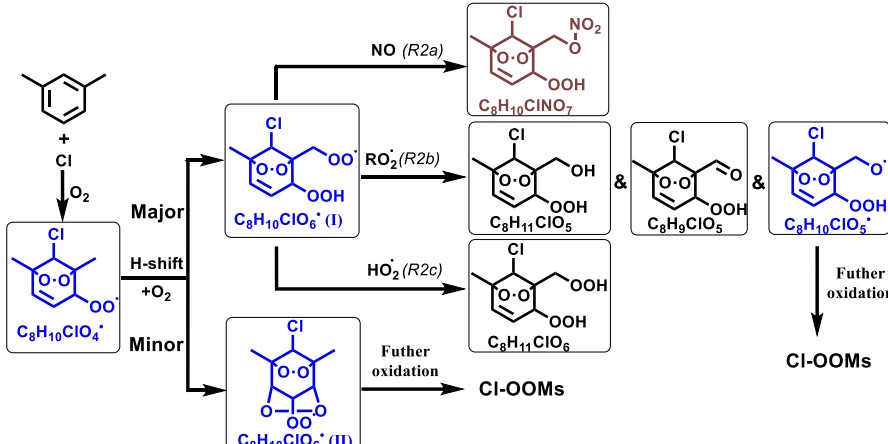

**Scheme 2.** Reaction pathways of the bicyclic peroxyl radical $C_8H_{10}ClO_4$ in the Cl-initiated reaction of m-xylene.
Blue, brown, and black formulae denote radicals, nitrogen-containing products, and Cl-OOMs products,
respectively. Radicals and stable products detected by nitrate-CI-APi-LToF are marked with black boxes.
**3.3.3. Dimer formation**
The accretion reaction ($RO_2 + R'O_2 \rightarrow ROOR' + O_2$) represents a pivotal source for dimer
compounds, originating from highly oxidized and functionalized $RO_2$ radicals (Ehn et al., 2014;
Zhao et al., 2018). As shown in Figure 2, $C_{16}H_{19}ClO_8$ and $C_{16}H_{19}ClO_{10}$ are two typical accretion
reaction products from reactions between Cl atoms and m-xylene without $NO_x$. In detail,
$C_{16}H_{19}ClO_{10}$ can be formed through the accretion reaction between $C_8H_{10}ClO_6\cdot$ and $C_8H_9O_6\cdot$
(Scheme 3). The formation pathways for $C_{16}H_{19}ClO_8$ are more varied compared to that of
$C_{16}H_{19}ClO_{10}$. It can be produced either via the accretion of a $C_8H_{10}ClO_6$ radical with a $C_8H_9O_4$
radical or through the reaction of a $C_8H_{10}ClO_4$ radical with a $C_8H_9O_6$ radical. Meanwhile,
$C_{16}H_{20}Cl_2O_{10}$, which is also detected during the reaction, can be formed via the accretion
reaction of two $C_8H_{10}ClO_6$ radicals.

**Scheme 3.** Accretion reaction pathways of peroxyl radical $C_8H_{10}ClO_6\cdot$ and $C_8H_9O_6\cdot$ in the Cl-initiated reaction of



m-xylene. Blue and black formulae denote radicals and stable products, respectively. Radicals and stable products
detected by nitrate-CI-APi-LToF are marked with black boxes.
Previous studies have reported dimer formation rates during the OH-initiated oxidation of
1,3,5-trimethylbenzene, ranging from $1.4 \times 10^{-10}$ to $25 \times 10^{-10}$ $cm^3$ molecule$^{-1}$ s$^{-1}$ (Berndt et al.,
2017). Compared to the OH reaction, Cl-RO$_2$ is produced in the Cl atom reaction, but its
reaction rate in the accretion reaction remains unknown. If Cl-RO$_2$ have lower reactivity (i.e.,
slower rate coefficients) for accretion reactions compared to non-Cl-containing RO$_2$, the same
generation rate of Cl-RO$_2$ and RO$_2$ would result in higher concentration of Cl-RO$_2$, resulting in
the high Cl-RO$_2$ signal detected. However, the generation rate of both Cl-RO$_2$ and RO$_2$ are
currently uncertain, further complicating the determination of the chemical mechanisms of Cl-
initiated reactions even more challenging.
**3.4. Cl-OOMs detection in suburban Shanghai.**
As shown in Figure 4A and Table S3, a total of 51 gaseous Cl-OOMs were identified during
winter in suburban Shanghai, whose high-resolution peak fittings measured by nitrate CI-APi-
LToF are shown in Figure S6. Figure S7 further demonstrates the accuracy of our peak
identification by showing the ratio of the fitted peak intensities versus the peak separation for
identified Cl-OOM peaks and adjacent ions from our ambient nitrate-CI-APi-LToF
measurements (Cubison and Jimenez, 2015). The peak separation, normalized to the half-width
at half-maximum ($\chi = \Delta t/HWHM$), is greater than 1 for all Cl-OOMs. Notably, 22% of the
peaks exhibit $\chi$ values between 1 and 2, indicating they are separable but closely spaced, while
78% of the Cl-OOMs ($\chi > 2$) are well-separated peaks.
Ambient Cl-OOMs consist of compounds mostly with a single Cl atom and only two with
two Cl atoms. The carbon and oxygen numbers of these Cl-OOMs ranged from 2 to 14 and 1
to 11, respectively. Figure 4B shows the abundance distribution of gaseous Cl-OOMs on the
basis of carbon and oxygen numbers in their molecular formulae. Approximately 80%
concentration of the identified Cl-containing molecules is C5-C9 Cl-OOMs, among which C6-
C9 Cl-OOMs represent a large fraction of ~79 %.





38 Cl-OOMs observed in field measurements were also identified in our laboratory
experiments, corresponding to reaction products of Cl atoms with toluene, m-xylene, or 1,2,4-
TMB. These results indicate that ambient C6-C9 Cl-OOMs exemplified by $C_7H_7N_2ClO_9$,
$C_8H_{10}NClO_7$, and $C_9H_{15}ClO_8$ are likely formed from Cl-initiated reactions with aromatic
compounds. On the other hand, isoprene was considered as a precursor of C5 Cl-OOMs (Breton
et al., 2018; Priestley et al., 2018; Wang and Ruiz, 2017).
Figure S8 shows the averaged diurnal variation of $C_7H_7N_2ClO_9$, $C_8H_{10}NClO_7$, and
$C_9H_{15}ClO_8$ during our field campaign with rainy days excluded. Similar to previous reports in
northern Europe and Beijing (Breton et al., 2018; Priestley et al., 2018), Cl-OOMs increased
with elevated solar radiation and their peaks appeared at around 12:00 p.m. (local time) hinting
that the formation of Cl-OOMs is connected with photochemistry. This suggests that while
$ClNO_2$ photolysis is a significant early morning source of Cl atoms, other sources such as the
photolysis of $Cl_2$, $ClONO_2$, HCl, ICl, and BrCl with sunlight can contribute to Cl atom
concentrations later in the day. (Peng et al., 2020)

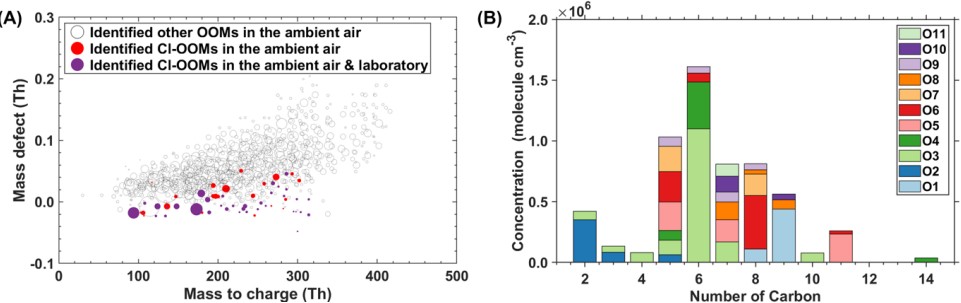


**Figure 4.** (A). Mass defect plot of detected OOMs by a nitrate-CI-APi-LToF in ambient air in suburban Shanghai.
Cl-OOMs only observed in the ambient atmosphere were marked in red, and Cl-OOMs observed in both ambient
and lab were marked in purple. The size of the circle is proportional to the concentration of compounds in the
ambient. (B). Distribution of carbon and oxygen number of gas-phase Cl-OOMs identified in suburban Shanghai.
The color codes correspond to the number of oxygen atoms.
**3.5. Health effect of ambient Cl-OOMs.**
Table S5 presents a detailed toxicity estimation of various Cl-OOMs, including $C_7H_7N_2ClO_9$,
$C_8H_{10}NClO_7$, and $C_9H_{15}ClO_8$, which were identified in the suburban Shanghai atmosphere and
laboratory experiments. These estimations are based on the potential chemical structures of





these Cl-OOMs. We characterized the toxicity of these Cl-OOMs using persistence,
bioaccumulative and toxic (PBT) criteria, as outlined in Table S5. Additionally, we conducted
a comparative analysis of their toxicity relative to that of naphthalene, a compound whose
atmospheric toxicity has been extensively studied.
Compounds with short OH radical reaction half-lives and bioconcentration factors (BCF) are
unlikely to be persistent and bioaccumulative. Among these Cl-OOMs, $C_8H_{10}NClO_7$ (II)
exhibits the slowest degradation rate with OH radicals with a half-life of 5 days, which suggests
that it is a potential new persistent pollutant (half-life > 2 days) in the atmosphere (Europe,

460   2023).

As indicated by the $pLD_{50}$ values, toxicity levels of most Cl-OOMs are lower than that of
naphthalene, except for $C_8H_{10}NClO_8$, $C_8H_{10}NClO_7$ (I), and $C_7H_7N_2ClO_8$ (Table S5). Among
these compounds, $C_8H_{10}NClO_7$ (I) is identified with the highest predicted toxicity, followed by
compound $C_8H_{10}NClO_8$. The $pLD_{50}$ of these compounds is categorized within level 3,
signifying their potential for a considerable acute toxicity.
Notably, the $pLD_{50}$ values of Cl-OOMs are found to be akin to those of naphthalene.
However, given that their concentration in suburban Shanghai (0.55 ppt) is merely one percent
of that of naphthalene (50 ppt) as measured in this study, the total probabilistic hazard quotient
(PrHQ, defined as the product of estimated human exposure (ambient concentration) and $pLD_{50}$
in this study) of Cl-OOMs is lower. Despite this lower PrHQ, it is important to recognize that
each of the evaluated Cl-OOMs may pose risks of developmental toxicity and mutagenicity,
which underscores the need for a thorough understanding of the toxicological implications of
Cl-OOMs in the atmosphere.
In summary, this study highlights the Cl-addition-initiated reaction as a non-negligible
pathway in the reaction of Cl atoms with aromatics. Field measurements in suburban Shanghai
revealed 51 gaseous Cl-OOM species, with 38 of these Cl-OOMs also detected in our laboratory
experiments. This suggests that these Cl-OOMs likely derive from reactions between Cl atoms
and aromatics. Considering the significant role of Cl atoms in daily atmospheric oxidation



processes, overlooking the Cl-addition pathway could lead to ignoring the formation of Cl-
OOMs from aromatic compounds in the atmosphere. This study, therefore, highlights the
necessity of incorporating both pathways in the models for a more accurate assessment of the
atmospheric fate of Cl atoms and aromatics in urban settings. In addition, health effect
evaluation indicates that all assessed Cl-OOMs may possess developmental toxicity, and nearly
half of the compounds may exhibit carcinogenic effects. Considering the critical role of
aromatics in the urban air and recent observations reporting increased levels of reactive chlorine
species in polluted atmospheres, our study offers timely insights into the chemical processes
between Cl atoms and aromatics occurring in anthropogenically influenced atmospheres and
the adverse health effects of these Cl-containing reaction products.
**Data availability.**
The data used in this study are available upon request from Lei Yao (lei_yao@fudan.edu.cn)
and Lin Wang (lin_wang@fudan.edu.cn).
**Author contributions.**
LY and LW conceived and designed this study and revised the manuscript. CL analyzed and
interpreted data, drafted and revised the manuscript. CL and YW contributed to the modeling
of the data. MF and XC contributed to the health effect analysis.
**Competing interests.**
The contact author has declared that none of the authors has any competing interests.
**Financial support.** This research was supported by the National Key Research and
Development Program of China (2022YFC3704100) and the National Natural Science
Foundation of China (21925601, 22376031, 92143301).



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
