# Peer review of "Formation of Chlorinated Organic Compounds from Cl Atom-Initiated"

_EGUsphere, 2025_

## Author Comment (AC1)

**RE: A point-to-point response to reviewers' comments**

Journal: *Atmospheric Chemistry and Physics*

Manuscript No.: egusphere-2025-607

Title: "Formation of Chlorinated Organic Compounds from Cl Atom-Initiated Reactions of Aromatics and Their Detection in Suburban Shanghai"

Author(s): Chuang Li, Lei Yao, Yuwei Wang, Mingliang Fang, Xiaojia Chen, Lihong Wang, Yueyang Li, Gan Yang, Lin Wang

Dear editor,

Thank you very much for providing us the opportunity to revise our manuscript. We appreciate reviewers' thorough evaluation and constructive comments, which have been invaluable when improving our manuscript.

We have observed the formation of Cl-containing oxygenated organic molecules (Cl-OOMs) from reactions of aromatics with Cl atoms in the laboratory. Our results revealed that Cl-addition plays a significant role in these reactions. Additionally, we detected 51 Cl-OOMs in the suburban atmospheres of Shanghai, 38 out of which were also observed in laboratory experiments. This finding provides new insights into atmospheric Cl-OOM formation. Moreover, we identified several toxic Cl-OOMs in the ambient air, underscoring their potential health impacts.

In the following response, we have addressed all reviewers' comments in detail and made necessary revisions to our manuscript. Key updates include the incorporation of new data from I-CIMS measurements, which revealed additional low-oxygenated radical species and provided further support for the Cl-addition pathway. In addition, the experiments with OH-scavenger were conducted to minimize interference from OH chemistry and underline the products from Cl-aromatic reactions. We believe the revisions address the comments thoroughly and further highlight the relevance of our findings to atmospheric chemistry. This work contributes to advancing the understanding of Cl-OOM formation and their implications in polluted urban environments.

The reviewers' comments are repeated in *italics* while the responses are in normal

fonts. All corrections made in the revised manuscript are marked in blue. The line numbers are those in a clean manuscript after all revisions have been accepted.

We are looking forward to your decision at your earliest convenience.

Sincerely,

Lei Yao

Lin Wang

**Reviewer #1:**

*Q0. The authors Li et al describe a series of laboratory flow tube studies and ambient urban measurements focused on Cl-radical initiated chemistry and resulting product Cl-OOMs. The results of the flow tube studies are potentially interesting and the ambient measurements are novel. However, I have major questions about the chemistry proposed within this work and how it may lead to the observed products. I believe the manuscript will only be suitable for publication with the addition of direct evidence in support of the proposed mechanisms.*

**Response:** We appreciate *Reviewer #1'* positive feedback and valuable suggestions, which have been taken into account in our revised manuscript.

*Major comments:*

*Q1a. My major issue is with the proposed mechanism for Cl addition to the aromatic rings. To my knowledge, this chemistry has not been directly supported through laboratory measurements, and in support of this mechanism the authors cite alone theoretical paper. While theory is undeniably useful in guiding mechanism development, I am hesitant to accept the Cl addition mechanism at the stated probability (14%) without direct measurements. Does the Cl-aromatic product analogous to the first-generation phenolic product from OH addition form? Are any other first-generation products observed that are analogous to those from established OH chemistry? Formation of smaller molecular fragmentation products should be common, given the high level of $RO_2$-$RO_2$ chemistry (Figure S10) and the expected fragmentation of ensuing RO radicals formed during some $RO_2$-$RO_2$ reactions. The Vocus-PTR should be well-suited to measuring at least some of these molecules. The authors do note indirect evidence for initial Cl addition in the lack of observation of potential second-generation radicals in the family $C_8H_{12}ClOx$ (Line 345). However, the authors also note that the detection of intermediate radicals is difficult (Line 285), and on its own this indirect evidence is not sufficiently convincing. I would suggest substantially deeper analysis of the data from the present experiments and ideally further*

*experiments more tailored specifically to detecting first-generation products of the Cl-aromatic reaction and constraining these reaction pathways.*

**Response:** We thank *Reviewer #1* for raising this important point regarding the need for direct experimental evidence to support the proposed Cl-addition mechanism. We carefully reanalyzed our data and conducted further experiments to probe the formation of first-generation products from the reactions of Cl-aromatics.

In these additional experiments, besides a nitrate-CIMS and a Vocus-PTR-LToF, an iodide-CIMS (I-CIMS) was also deployed to broaden the detection coverage of first-generation products, particularly for RO and $RO_2$ radicals, and less oxygenated species.

From our original experiments of Cl radicals and m-xylene, the spectrum peaks corresponding to $C_8H_{10}O$ detected by Vocus-PTR-LToF and $C_8H_{12}O_5$ detected by nitrate-CI-APi-LToF were identified. These two compounds are consistent with first-generation products analogous to those formed via OH addition to aromatic rings (OH-initiated reactions) or through subsequent oxidation following H-substitution on the ring (Cl-initiated reactions) (refer to Scheme R1). Since both OH- and Cl-initiated reactions of m-xylene can produce $C_8H_{10}O$ and $C_8H_{12}O_5$, the presence of these first-generation products alone does not allow us to distinguish whether OH chemistry is active under our experimental conditions.

**Scheme R1.** Proposed reaction mechanisms of first-generation products from Cl-initiated and OH-initiated reactions of m-xylene. Blue and black formulae denote radicals and stable products, respectively.

Therefore, to address this concern, we conducted additional experiments with 20 ppm of CO introduced as an OH scavenger, aiming to ensure most of the OH radicals were scavenged and suppress OH-related oxidation. Upon CO addition, the signal intensity of $C_8H_{12}O_5$ showed a noticeable decrease but did not fall to baseline (Figure R1(a)). In contrast, the first-generation products from Cl-initiated reactions—namely $C_8H_8O$ and $C_8H_{11}ClO_2$ (formation mechanisms refer to Scheme R2) —were unaffected by CO addition. The signal intensity of another Cl-initiated product, $C_8H_{11}ClO_4$ (formation mechanisms refer to Scheme R2), decreased by approximately 50% upon CO addition. This indicates that its formation likely involves OH chemistry in addition to Cl reactions.

[Figure]

**Figure R1.** Time-resolved signal intensities of radicals and selected first-generation products from the Cl + m-xylene reaction in the absence of NOx. (a) Normalized signals of representative first-generation products, categorized by their dominant formation pathway (Cl-initiated vs. OH-initiated). $C_8H_8O$ and $C_8H_{10}O$ were detected by Vocus-PTR-LToF; $C_8H_{11}ClO_2$, $C_8H_{11}ClO_4$, and $C_8H_{12}O_3$ were detected by I-CIMS; $C_8H_{12}O_5$ was detected by nitrate-CI-APi-LToF. (b) Signals of the first-generation radicals of $C_8H_{10}ClO_{2-3}$ detected by nitrate-CI-APi-LToF and $C_8H_{10}ClO_{4-6}$ detected by I-CIMS. The vertical dashed lines indicate the time points when UVA light was turned on (t = 0 min) and when CO was introduced (t $\approx$ 27 min) to suppress OH reactions.

**Scheme R2.** Proposed reaction mechanisms of m-xylene with Cl atoms leading to the formation of Cl-OOMs and non-Cl-OOMs, Blue and black formulae denote radicals and stable products, respectively.

In our additional experiments, I-CIMS was also employed to directly detect $C_8H_{11}O_{2-3}$ radicals (formation mechanisms refer to Scheme R2) detected as $C_8H_{11}O_{2-3} \cdot I^-$ ions. Although quantitative analysis of these radicals is challenging due to their high reactivity and uncertain ionization efficiency, the detection of these first-generation radical products provides further support for the formation of first-generation products from the Cl-aromatic reaction. As shown in Figure R1 (b), the signals of $C_8H_{11}O_{2-6}$ radicals (Scheme R2) remained unchanged upon CO addition, further confirming that their formation from Cl-aromatic reactions, instead of OH-aromatic reactions. Collectively, these results affirm the minor contribution of OH reactions to product formation without altering the main mechanistic conclusion that Cl-addition remains the dominant pathway for chlorinated radical and OOM formation in this system.

In light of this, we have revised the relevant sentence in Supporting Information, which reads (SI, Text S5):

"Although previous studies using modeling approaches have suggested that OH radicals are not formed in the Cl-atom-initiated oxidation of ethylbenzene under NOx-free conditions, OH can still be generated through H-abstraction from methyl groups by $HO_2$ (Bhattacharyya et al., 2023). Therefore, OH chemistry may also occur in our

low-NOx experiments. To assess the potential influence of OH on product formation, we conducted additional experiments with the introduction of 20 ppm CO as an OH scavenger, following the setup described in Table 1 (Experiment 6). Product signals were measured using Vocus-PTR-LToF, I-CIMS, and nitrate-CI-APi-LToF.

Figure S12 displays the real-time signals of the first-generation products of Cl-initiated reaction and OH-initiated reaction and $C_8H_{11}ClO_{2-6}$ radicals. Upon CO (~20 ppm) addition to ensure most of OH radicals were scavenged, the signal intensity of $C_8H_{12}O_5$ showed a noticeable decrease but did not fall to baseline (Figure S12a). In contrast, the first-generation products from Cl-initiated reactions—namely $C_8H_8O$ and $C_8H_{11}ClO_2$—were unaffected by CO addition. The signal intensity of another Cl-initiated product, $C_8H_{11}ClO_4$, decreased by approximately 50% upon CO addition. This indicates that its formation likely involves OH chemistry in addition to Cl reactions. These changes in stable products demonstrate the presence of OH chemistry even under low NOx conditions, suggesting that OH chemistry contributes to the formation of some of highly oxygenated Cl-OOMs, thereby enhancing their yield.

Meanwhile, the $C_8H_{11}ClO_{1-6}$ radical signals (Figure S12b) remained unaffected by CO, confirming that these radicals originate exclusively from Cl-initiated chemistry. Collectively, these results affirm the contribution of OH to product formation without altering the main mechanistic conclusion that Cl-addition remains the dominant pathway for chlorinated radical and OOM formation in this system."

*Q1b. Relatedly, I do not believe the mechanism illustrated in the upper half of scheme 1 describing OOM formation following H-abstraction by Cl is reasonable. H-abstraction is expected to occur on the methyl substituents, and the internal RO2-H migrations described following initial H-abstraction are not supported by prior literature. To my knowledge, internal H-migration of an aromatic H is not expected to be a reasonable pathway. Additionally, though internal H-migration across an aromatic ring hasn't been studied (to my knowledge), a 1,7-migration between primary carbons is predicted to be slow (Vereecken and Noziere, 2020). If this were to occur, the second H-migration to form the radical C8H9O4 would be expected to immediately*

*collapse to form an aldehyde in the closed shell molecule C8H8O3 and an OH radical (Bianchi et al., 2019). This suggests other formation pathways for the observed non-Cl containing radicals and closed-shell products. The simplest explanation would be OH addition chemistry, with OH forming from HO2 through H-abstraction at the methyl groups (see, e.g., the already cited Bhattacharyya et al., 2023). These observations call into question the authors' statements on a lack of OH chemistry in NOx-free experiments (line 360-361), with further implications for the potential mechanisms by which Cl-OOMs may form.*

**Response:** We appreciate the reviewer's helpful suggestions about the proposed H-abstraction and autoxidation mechanism shown in the upper part of Scheme 1.

As mentioned in our response to Q1a, we conducted additional experiments using 20 ppm CO as an OH scavenger to minimize the influence of OH-initiated chemistry. The results showed that while certain OH-derived products were suppressed, the compounds such as $C_8H_{10}O$ and $C_8H_{12}O_5$ remained detectable. This indicates that Cl-initiated oxidation alone can also lead to the formation of similar products (Scheme R1).

We acknowledge that the 1,7-hydrogen shift between primary carbons was indeed predicted to be slow and that internal H-migration of an aromatic hydrogen is not generally considered a favorable pathway. In light of these valid concerns, we have revised Scheme 1 in the manuscript. The updated scheme 1 (also shown in Scheme R3) now focuses primarily on the Cl-addition pathway, which is more consistent with experimental evidence and mechanistic plausibility. Furthermore, we have expanded the mechanistic discussion in the Supplementary Information to include potential H-substitution pathways. The specific H-substitution reaction pathways are illustrated in Scheme R4, where Cl atoms can abstract hydrogen atoms not only from the methyl groups of m-xylene, but also from aromatic ring positions, followed by autoxidation to form non-chlorinated OOMs.

**Scheme R3.** Proposed reaction mechanisms of m-xylene with Cl atoms leading to the formation of Cl-OOMs. Blue and black formulae denote radicals, and stable products, respectively. Radicals and stable products detected by nitrate-CI-APi-LToF are marked with black boxes.

**Scheme R4.** Proposed reaction pathways for the formation of non-chlorinated oxygenated organic molecules (OOMs) from the Cl-initiated oxidation of m-xylene. The scheme illustrates two primary mechanisms: (1) H-abstraction by Cl atoms from the methyl substituents, followed by $O_2$ addition and autoxidation; and (2) H-abstraction from aromatic positions, also followed by autoxidation. Both pathways can lead to the formation of a range of non-Cl-containing $RO_2$ radicals and closed-shell products. Representative detected or inferred molecular formulas are labeled in blue.

We have revised the relevant section of the manuscript as follows (Lines 341–353):

"The initial reaction of m-xylene ($C_8H_{10}$) with Cl atoms can occur through three pathways: the Cl-addition pathway, leading to the formation of a $C_8H_{10}Cl$ radical ($C_8H_{10}Cl\cdot$, refer to Scheme 1) or the H-abstraction pathway, forming a $C_8H_9$ radical (refer to Scheme S1) and $C_8H_9O_2$ radical (refer to Scheme S1). Then, both $C_8H_9O_2\cdot$ and $C_8H_{10}Cl\cdot$ can in turn undergo autoxidation via the formal addition of $O_2$ to produce peroxy radicals of $C_8H_{11}O_5\cdot$or $C_8H_{10}ClO_6\cdot$ (Vereecken and Nozière, 2020; Bianchi et al., 2019). The peroxyl radical $C_8H_{10}ClO_6\cdot$was identified as the predominant species in terms of signal (Table S1). Although these Cl-OOMs-to-non-Cl-OOMs signal ratios may not accurately represent their relative concentrations due to sensitivity differences of these radicals towards the reagent ions (($HNO_3)_{0-1}\cdot NO_3^-$), it is still noteworthy that these Cl-$RO_2$ overlooked in previous studies were directly observed in such a reaction system, thereby suggesting that Cl-addition pathway is indeed present in the initial reaction steps of reactions between Cl atoms and aromatics."

*Minor comments:*

*Q2. Line 365: More reduced molecules could also form through multi-generation chemistry when H-abstraction occurs from a carbon with an OH or OOH substituent, leading to the formation of a carbonyl.*

**Response:** We appreciate the reviewer's helpful comment. We agree that more reduced products, such as carbonyl-containing compounds, could indeed form through multi-generation oxidation steps. In light of this, we have revised the relevant sentence in the manuscript, which reads (Line 379-385):

"Moreover, under high $NO_x$ conditions in our experiments, the dominant Cl-OOM $C_8H_{10}ClNO_7$ (Scheme S2) contains two fewer H atoms than the $C_8H_{12}ClNO_8$ proposed by Jahn et al. (2024) as a secondary Cl-addition product (see Scheme S3). While this difference might suggest the presence of an additional double bond or a ring in C8H10ClNO7, it could also result from multi-generation oxidation chemistry. Specifically, H-abstraction from a carbon bearing an OH or OOH can form a carbonyl, thereby reducing the hydrogen count without requiring a new double bond."

*Q3. Lin 443, Figure 4: I would like to see more discussion on some of the higher-concentration compounds, specifically C2O2, C6O3/4, C8O5, and C9O1. Even if these compounds were not observed in the flow tube studies, the observation is both novel and useful and can provide further insight on ambient Cl chemistry and/or primary Cl-OVOC/Cl-OOM emissions.*

**Response:** We thank the reviewer for this insightful comment. We add a more detailed discussion of the higher-abundance compounds observed in ambient air, including $C_2H_2Cl_2O_2$, $C_6H_4ClNO_{3-4}$, $C_8H_7ClO_6$, and $C_9H_6ClNO$ in the revised manuscript, which reads (Lines 456-466):

"Several relatively abundant ambient Cl-OOMs, specifically $C_2H_2Cl_2O_2$, $C_6H_4ClNO_{3-4}$, $C_8H_7ClO_6$, and $C_9H_6ClNO$, were detected in the field measurements (Figure 4A). The presence of $C_2H_2Cl_2O_2$ suggests possible contributions from the oxidation or atmospheric degradation of chlorinated solvents, combustion emissions, or industrial processes (Wang et al., 2021). The $C_6H_4ClNO_{3-4}$ may originate from Clinduced oxidation of aromatics. Similarly, $C_8H_7ClO_6$ and $C_9H_6ClNO$ could be derived from more oxidized aromatic compounds undergoing multi-step oxidation or fragmentation. The observation of these species highlights the complexity and diversity of Cl chemistry in suburban environments, underscoring the importance of further studies integrating broader precursor sets, longer aging times, and additional reaction pathways to better understand their sources and formation mechanisms."

*Q4. Line 459: fix citation.*

**Response:** Revised as suggested.

*Q5. Line 474: label concluding section.*

**Response:** Revised as suggested.

*Q6. References not already within text.*

*Vereecken, L., & Nozière, B. (2020). H migration in peroxy radicals under atmospheric conditions. Atmospheric Chemistry and Physics, 20(12), 7429–7458. [https://doi.org/10.5194/acp-20-7429-2020](https://doi.org/10.5194/acp-20-7429-2020)*

**Response:** This reference was added to the revised manuscript (Line 345-347), which reads,

"Then, both $C_8H_9O_2\cdot$ and $C_8H_{10}Cl\cdot$ can in turn undergo autoxidation via the H-shift and addition of $O_2$ to produce peroxy radicals of $C_8H_{10}O_5\cdot$ or $C_8H_{10}ClO_6\cdot$ (Vereecken and Nozière, 2020; Bianchi et al., 2019)."

**References**

Bhattacharyya, N., Modi, M., Jahn, L. G., and Ruiz, L. H.: Different chlorine and hydroxyl radical environments impact m -xylene oxidation products, Environ. Sci.: Atmos., https://doi.org/10.1039/d3ea00024a, 2023.

Bianchi, F., Kurtén, T., Riva, M., Mohr, C., Rissanen, M. P., Roldin, P., Berndt, T., Crounse, J. D., Wennberg, P. O., Mentel, T. F., Wildt, J., Junninen, H., Jokinen, T., Kulmala, M., Worsnop, D. R., Thornton, J. A., Donahue, N., Kjaergaard, H. G., and Ehn, M.: Highly Oxygenated Organic Molecules (HOM) from Gas-Phase Autoxidation Involving Peroxy Radicals: A Key Contributor to Atmospheric Aerosol, Chem Rev, 119, 3472–3509, https://doi.org/10.1021/acs.chemrev.8b00395, 2019.

Jahn, L. G., McPherson, K. N., and Ruiz, L. H.: Effects of Relative Humidity and Photoaging on the Formation, Composition, and Aging of Ethylbenzene SOA: Insights from Chamber Experiments on Chlorine Radical-Initiated Oxidation of Ethylbenzene, ACS Earth Space Chem., 8, 675–688, https://doi.org/10.1021/acsearthspacechem.3c00279, 2024.

Vereecken, L. and Nozière, B.: H migration in peroxy radicals under atmospheric conditions, Atmos. Chem. Phys., 20, 7429–7458, https://doi.org/10.5194/acp-20-7429-2020, 2020.

Wang, M., Yang, L., Liu, X., Wang, Z., Liu, G., and Zheng, M.: Hexachlorobutadiene emissions from typical chemical plants, Front. Environ. Sci. Eng., 15, 60, https://doi.org/ 10. 1007 /s11783-020-1352-8, 2021.

**Reviewer #2:**

*Q0. This manuscript presents a detailed and well-structured investigation into the formation mechanisms of chlorine-containing oxygenated organic molecules (Cl-OOMs) from Cl-initiated reactions with aromatics in both laboratory experiments and ambient measurements. The study provides direct evidence of Cl-addition reactions and explores the role of NOx in modulating Cl-OOM formation. The identification of 51 Cl-OOMs in suburban Shanghai, with 38 also detected in the laboratory, strengthens the claim that Cl-initiated oxidation of aromatics is an important atmospheric process. Furthermore, the toxicity evaluation adds valuable insights into the potential health risks associated with these compounds. Overall, the study is comprehensive and provides significant contributions to the field. However, several points require further clarification or additional discussion to strengthen the manuscript.*

**Response:** We appreciate *Reviewer #2*' positive feedback and valuable suggestions.

*Q1a. Line 121 (experimental setup): The mixing of a few sccm of gas (2.5 sccm) with a 10 lpm flow may lead to incomplete mixing or loss of reactants. Have the authors verified that the minor flow contributions are fully entrained and do not experience losses?*

**Response:** To ensure proper entrainment and verify the uniform mixing of the trace gases, we routinely monitored the concentrations of precursors and NOx prior to each experiment. Specifically, the concentrations of gas-phase precursors were measured using a Vocus-PTR-LToF, and $NO+NO_2$ concentrations were quantified using a NOx analyzer. The measured concentrations remained stable across repeated measurements, indicating good mixing. Even if there were minor losses before reaching the reaction zone, these losses appeared to be constant, suggesting that the concentrations measured at the outlet reliably represent the actual concentrations inside the flow tube.

*Q1b.Additionally, is nitrate-CI-APi-LToF sampling from the center of the flow tube reactor while other instruments sample from different radial positions? If so, have the*

*authors considered the potential radial inhomogeneity in reactions, especially in cases where reactants are not fully mixed before entering the flow tube?*

**Response:** We appreciate the reviewer's suggestion regarding the potential for radial inhomogeneity in the flow tube reactor and differences in sampling positions for various instruments. A schematic illustration of the downstream sampling configuration is shown in Figure R2, highlighting the positions of all measurement instruments relative to the flow tube outlet. In details, the nitrate-CI-APi-LToF samples from the centerline of the flow, while the other instruments sample from side ports located 4 cm off-center. To minimize spatial bias, these side sampling inlets were carefully aligned and directed toward the center of the flow tube. Our flow tube is 120 cm in length and has an internal diameter of 8 cm. Given this relatively small diameter and the laminar flow conditions (Reynolds number $\approx$ 200), radial mixing is expected to be efficient over the residence time of ~36 s. Although slight concentration gradients between the center and edge may exist, the small cross-sectional dimensions ensure that such differences are minimal. Moreover, the side sampling ports are oriented toward the centerline of the flow tube to further reduce spatial bias.

[Figure]

**Figure R2.** Schematic of the sampling setup at the outlet of the flow tube reactor. All sampling inlets were positioned on the same downstream face of the flow tube to ensure consistency in sampling location and minimize spatial bias.

We have added this clarification to the revised Supplementary Information, describing the flow tube geometry and radial sampling considerations, which reads (SI, Figure S1).

"To minimize potential spatial discrepancies, care was taken to ensure consistent sampling across instruments. In our flow tube system (120 cm length, 8 cm internal diameter), the nitrate-CI-APi-LToF samples from the center of the flow tube, while other instruments (e.g., Vocus-PTR-LToF, I-CIMS, NOx monitor, SMPS, and PSM) sample from side ports positioned 4 cm off-center—i.e., halfway across the tube radius. Under the typical laminar flow conditions in our system (Reynolds number ≈ 200) and a residence time of ~36 s, radial diffusion is sufficient to promote near-uniform mixing across the tube cross-section. Furthermore, the side sampling ports are directed toward the centerline to minimize radial bias. Based on the geometry and previous test comparisons, the differences in concentration between sampling positions were determined to be negligible."

*Q2. Line 127: How was the VOC introduced into the system specifically? Was it through a permeation tube or another method? A clearer description of the VOC introduction process would be beneficial.*

**Response:** In our experiments, we used a custom-made gas cylinder prepared using a VOC pressure-dividing system. The process began by evacuating a clean stainless-steel cylinder to ~$10^{-2}$ mbar. A small amount of solid VOC sample, precooled by liquid nitrogen, was then placed in the vacuum system. Upon removal of the liquid nitrogen, the VOC was allowed to volatilize and fill the evacuated cylinder. The pressure increase was monitored using a high-precision pressure gauge. Once the desired partial pressure was achieved, the VOC inlet valve was closed, and the cylinder was pressurized to 70 psi using high-purity nitrogen gas.

To ensure accurate VOC concentrations, the custom-made gas cylinder was first validated using a Vocus-PTR-LToF, which had been calibrated on the same day using certified commercial gas standards (Air Liquide Co., Ltd.). These standards included toluene, m-xylene, acetonitrile, acetaldehyde, methyl ethyl ketone (MEK), acetone, and acrylonitrile. During our experiments, the aromatics were controlled using a mass flow controller (MFC), with flow rates typically ranging from 0.05 to 10 sccm, depending on the compound and experimental condition.

This clarification has been added to the revised Supporting Information to provide a more comprehensive description of the VOC introduction procedure, as outlined in Text S1 of the Supporting Information.

"Precursors (aromatics) were introduced into the flow tube using a custom-prepared gas cylinder generated through a low-pressure VOC loading system. Briefly, a clean stainless-steel cylinder was evacuated to $\sim 10^{-2}$ mbar, after which a small amount of solid-phase aromatics precooled by liquid nitrogen was allowed to volatilize and fill the cylinder. The resulting pressure increase was monitored using a precision gauge, and once the target partial pressure was reached, the cylinder was sealed and then pressurized to 70 psi with high-purity nitrogen gas. To verify the precursor concentration, each gas cylinder was validated using a Vocus-PTR-LToF calibrated on the same day with certified commercial standards (Air Liquide Co., Ltd.). These VOCs standards included toluene, m-xylene, acetonitrile, acetaldehyde, methyl ethyl ketone (MEK), acetone, and acrylonitrile. VOCs were introduced into the flow system through a mass flow controller (MFC), with flow rates ranging from 0.05 to 10 sccm, depending on the specific compound and experimental conditions."

*Q3. Lines 135-140: The consistency of Cl atom concentrations across experiments with different VOCs needs further support. Have the authors confirmed that Cl atom concentrations remain consistent for the three VOCs under the same flow conditions? Providing a graphical representation (e.g., VOC concentration versus time) would strengthen this claim.*

**Response:** We appreciate the reviewer's comment regarding the consistency of Cl atom concentrations across experiments with different VOCs. Figure R3 shows the decay of aromatic precursors in the experiments in the absence of NOx. In our study, the concentration of Cl atoms was not directly measured but was obtained based on the decay of each VOC precursor using Eq. (1), with literature-based rate coefficients for Cl + VOC reactions (Wang et al., 2005). Although the three aromatic precursors in Figure R3 exhibit different degrees of concentration decay, these differences are attributed to variations in their reaction rate constants with Cl atoms. Under identical

flow and illumination conditions, the decay rates of toluene, m-xylene, and 1,2,4-TMB were used to calculate the Cl atom concentration, and the estimated Cl exposures remained within the range of $(1.2\text{-}2.0) \times 10^9$ molecules cm$^{-3}$ s.

$$[Cl] = -1/kt \times \ln([Aromatics]_t/[Aromatics]_0) \qquad \text{Eq. (1)}$$

where $[Aromatics]_0$ and $[Aromatics]_t$ are the initial concentration and the concentration after a reaction time $t$ of aromatic precursors, respectively. $[Cl]$ is the estimated concentration of Cl atoms in the flow tube. In our flow tube experiments, the extent of oxidation is quantified using the parameter of Cl exposure, defined as $[Cl]$ multiplied by the reaction time $t$.

[Figure]

**Figure R3.** Time profile of aromatic precursors (toluene, m-xylene, and 1,2,4-trimethylbenzene) measured by Vocus-PTR-LToF during the Cl-initiated oxidation experiment (refer to Exp. 1,5, and 9 in Table 1). The black dashed line at t = 0 marks the onset of UVA illumination, which initiates the photolysis of Cl$_2$ and the subsequent formation of Cl radicals.

A graphical representation of VOC concentration versus time has now been added as Figure S2 to the Supporting Information to support this claim. Corresponding descriptions have also been added to the revised manuscript to enhance clarity, which reads (Line 137-141),

"The concentration of Cl atoms was controlled by adjusting the flow rate of Cl$_2$. The mean concentrations of Cl atoms were determined according to the decay of aromatic precursors (Figure S2) and calculated using Eq.(1), with reaction rate coefficients $k$ of $6.2\times10^{-11}$ cm$^3$ molecule$^{-1}$ s$^{-1}$, $1.35\times10^{-10}$ cm$^3$ molecule$^{-1}$ s$^{-1}$, and

2.42×10$^{-10}$ cm$^3$ molecule$^{-1}$ s$^{-1}$ for reactions between Cl atoms and toluene, m-xylene, and 1,2,4-TMB, respectively ".

*Q4. Lines 234-242: When NOx was introduced, both non-Cl-OOMs and Cl-OOMs increased in molar yield, but the explanation focuses mainly on OH chemistry. Given that Cl-OOMs also increased, further discussion is needed on why this occurs.*

**Response:** Indeed, our initial explanation focused primarily on the enhancement of non-Cl-OOMs via OH chemistry. In the revised manuscript, we have expanded this discussion to better address the Cl-OOM trend as well.

Specifically, we now note that, in addition to OH-driven pathways, the increase in Cl-OOM yields may also result from changes in $RO_2$ termination chemistry. Under $NO_x$ conditions, the suppression from $RO_2$–NOx reactions reduced the formation of dimers or multimers, shifting the product distribution toward monomeric species. This shift can enhance the apparent yield of monomeric Cl-OOMs, which are the species primarily quantified in our study. Furthermore, NO-promoted $RO_2$ termination may stabilize Cl-containing intermediates, facilitating Cl-OOM formation.

Additionally, high $NO_x$ conditions promote the formation of a wider variety of nitrogen-containing OOMs. Given the varying detection sensitivities of the nitrate-CI-APi-LToF toward different nitrogenated species, this compositional complexity may introduce additional uncertainty in yield quantification. These considerations have been added to the revised manuscript to provide a more comprehensive interpretation, which reads (Lines 248–254).

"In addition, the increased Cl-OOM yields under $NO_x$ conditions may result from the suppression of dimer or multimer formation (Table 1), which shifts the product distribution toward monomeric Cl-OOMs and thus leads to potential higher apparent yields. Moreover, $NO_x$-promoted chemistry facilitates the formation of nitrogen-containing OOMs with diverse structures and functionalities. The nitrate-CI-APi-LToF exhibits different detection sensitivities toward these species, which may also influence the estimated molar yields under high $NO_x$ conditions."

*Q5. Lines 423-424: The manuscript mentions measurements of Cl-OOMs with one oxygen atom using nitrate-Cl-APi-LTOF. How confident are the authors in detecting these low-oxygenated species using nitrate-Cl-APi-LTOF? Additional justification for measurement accuracy would be helpful.*

**Response:** We thank the reviewer for this valuable comment. We acknowledge that the detection of low-oxygenated Cl-containing organic molecules (e.g., those with only one oxygen atom) by nitrate-Cl-APi-LTOF involves a degree of uncertainty, as nitrate ion ($NO_3^-$) clustering is generally more efficient toward highly oxygenated or acidic molecules.

However, prior studies (e.g., Bianchi et al., 2019; Ehn et al., 2014) have shown that nitrate-Cl-APi-LTOF can detect certain moderately oxygenated species if they contain functional groups with sufficient gas-phase acidity or hydrogen bonding ability (e.g., hydroxyl, carboxyl, or halogenated moieties), which enhance clustering efficiency.

We have clarified these limitations in the Supporting Information and now include the following text, which (legend for Figure S3) reads.

"While the detection of highly oxygenated organic molecules by nitrate-CI-APi-LToF is well-established, the identification of species with only one or two oxygen atoms (e.g., $C_8H_{12}Cl_2O$ or $C_2H_2Cl_2O_2$) involves greater uncertainty due to lower ionization efficiency. Nevertheless, previous studies have shown that nitrate-CI-APi-LToF can detect some moderately oxygenated compounds if they contain functional groups with sufficient gas-phase acidity or hydrogen bonding capacity (e.g., hydroxyl, carboxyl, or halogen substituents), which enhance clustering efficiency (Bianchi et al., 2019; Ehn et al., 2014)."

*Q6. Lines 434-438: While 38 species were observed in both laboratory and field settings, only three were selected for diurnal variation analysis. What was the rationale for this selection? Do the remaining 35 species exhibit similar daily trends? A discussion or additional data supporting the selection of these three species would enhance clarity.*

**Response:** We have now provided a broader overview of the diurnal patterns of all 51

Cl-OOMs detected in ambient air. These trends are now presented in the newly added Figure R4 (also shown as Figure S9 in the revised Supplementary Information), and the corresponding molecular formulas are listed in Table R1 (Table S3 in the revised Supplementary Information).

The three compounds selected for detailed diurnal variation analysis were chosen because their molecular formulas ($C_7H_7ClN_2O_9$, $C_8H_{10}ClNO_7$, and $C_9H_{15}ClNO_8$) are representative of highly oxygenated Cl-OOMs that could plausibly form via well-defined Cl-initiated mechanisms. These species were selected due to their relatively high signal intensities and the availability of proposed formation pathways, which allowed us to assign structures tentatively. These assignments are critical for later assessments of atmospheric reactivity and potential health impacts. To provide a more comprehensive discussion, we included the real-time profiles of all 51 Cl-OOMs and analyzed the possible reasons behind their differing temporal behaviors. We have clarified this rationale in the revised manuscript (Lines 467-473), which reads.

"The diurnal profiles of all 51 Cl-OOMs identified in the ambient were shown in Figure S9. Similar to previous reports in northern Europe and Beijing (Breton et al., 2018; Priestley et al., 2018), most Cl-OOMs increased with elevated solar radiation and their peaks appeared at around 12:00 p.m. (local time) hinting that the formation of Cl-OOMs is connected with photochemistry. This suggests that while $ClNO_2$ photolysis is a significant source of Cl atoms in the early morning, and other sources such as the photolysis of $Cl_2$, $ClONO_2$, HCl, ICl, and BrCl by sunlight can also contribute to Cl atom concentrations later in the day (Peng et al., 2020)."

[Figure]

**Figure R4.** Mean diurnal profiles of all 51 Cl-OOMs detected by nitrate-CI-APi-LToF from December 14th, 2022, to February 2nd, 2023. All Cl-OOMs listed in Table R1.

**Table R1.** Average concentrations of ambient Cl-OOMs detected by nitrate-Cl-APi-LTOF between December 14th, 2022, and February 2nd, 2023, at the Dianshan Lake (DSL) Air Quality Monitoring Supersite in Shanghai, China.

| Molecular formula | Concentration* ($cm^{-3}$) | Molecular formula | Concentration * ($cm^{-3}$) |
|---|---|---|---|
| $C_2H_2Cl_2O_2$ | $2.6\times10^5$ | $C_7H_6NClO_3$ | $1.7\times10^5$ |
| $C_2H_3ClO_2$ | $9.0\times10^4$ | $C_7H_6N_3ClO_8$ | $5.4\times10^4$ |
| $C_2H_9ClO_3$ | $7.2\times10^4$ | $C_7H_7N_2ClO_8$ | $3.6\times10^4$ |
| $C_3H_3ClO_2$ | $8.3\times10^4$ | $C_7H_7N_2ClO_9$ | $2.6\times10^4$ |
| $C_3H_3ClO_3$ | $5.1\times10^4$ | $C_7H_7N_2ClO_{10}$ | $1.3\times10^5$ |
| $C_4H_3ClO_3$ | $4.9\times10^4$ | $C_7H_9ClO_{11}$ | $5.7\times10^4$ |
| $C_4H_5ClO_3$ | $3.2\times10^4$ | $C_7H_{11}N_2ClO_8$ | $5.7\times10^4$ |
| $C_5H_5ClO_3$ | $1.2\times10^5$ | $C_7H_{13}ClO_9$ | $5.5\times10^4$ |
| $C_5H_5ClO_5$ | $9.4\times10^4$ | $C_8H_7ClO_6$ | $4.4\times10^5$ |
| $C_5H_6NClO_2$ | $6.3\times10^4$ | $C_8H_7ClO_7$ | $8.2\times10^4$ |
| $C_5H_6NClO_6$ | $1.3\times10^5$ | $C_8H_{10}NClO_7$ | $2.1\times10^4$ |
| $C_5H_6NClO_7$ | $7.0\times10^4$ | $C_8H_{10}NClO_8$ | $3.4\times10^4$ |
| $C_5H_7ClO_9$ | $7.4\times10^4$ | $C_8H_{10}NClO_9$ | $5.1\times10^4$ |
| $C_5H_7N_2ClO_4$ | $8.0\times10^4$ | $C_8H_{12}NClO_7$ | $7.5\times10^4$ |
| $C_5H_8NClO_5$ | $1.4\times10^5$ | $C_8H_{12}Cl_2O$ | $1.1\times10^5$ |
| $C_5H_9ClO_6$ | $1.2\times10^5$ | $C_9H_6NClO$ | $4.4\times10^5$ |
| $C_5H_9N_2ClO_7$ | $1.4\times10^5$ | $C_9H_{11}ClO_{10}$ | $4.7\times10^4$ |
| $C_6H_4NClO_3$ | $1.1\times10^6$ | $C_9H_{15}ClO_8$ | $7.5\times10^4$ |
| $C_6H_4NClO_4$ | $3.0\times10^5$ | $C_{10}H_{12}NClO_3$ | $7.7\times10^4$ |
| $C_6H_5N_2ClO_4$ | $2.2\times10^4$ | $C_{11}H_6NClO_5$ | $8.0\times10^4$ |
| $C_6H_5N_2ClO_6$ | $7.1\times10^4$ | $C_{11}H_7ClO_5$ | $6.7\times10^4$ |
| $C_6H_7N_2ClO_4$ | $6.4\times10^4$ | $C_{11}H_8NClO_6$ | $2.7\times10^4$ |
| $C_6H_9ClO_9$ | $5.3\times10^4$ | $C_{11}H_{11}ClO_5$ | $8.7\times10^4$ |
| $C_7H_5ClO_{11}$ | $4.2\times10^4$ | $C_{14}H_{12}NClO_4$ | $3.6\times10^4$ |
| $C_7H_5N_2ClO_5$ | $1.2\times10^5$ | $C_{16}H_{11}ClO_4$ | $1.0\times10^5$ |
| $C_7H_6NClO_5$ | $6.2\times10^4$ | | |

**\*** Quantification of Cl-OOMs using the calibration factor of sulfuric acid may result in an uncertainty ($\pm$ 50% as that of sulfuric acid) in the Cl-OOMs concentrations. The detection limit of the nitrate-Cl-APi-LTOF is $1.4\times10^4\,cm^{-3}$.

*Q7. Lines 436-441: Cl Radical Concentrations at Noon: The manuscript suggests that Cl radicals peak at noon, but many studies report higher morning concentrations due to ClNO2 photolysis. What evidence supports a midday peak at this measurement site in this study? If none exists, could these Cl-containing compounds be formed from Cl-VOC reactions involving OH rather than Cl oxidation? Providing further discussion or alternative hypotheses would improve the robustness of this conclusion.*

**Response:** We thank the reviewer for this insightful comment. We fully agree that many previous studies have reported elevated Cl atom concentrations in the early morning, primarily attributed to the rapid photolysis of ClNO$_2$ shortly after sunrise. However, in our study, the observed diurnal profiles of Cl-OOMs exhibit distinct peaks around noon (Figure R4), rather than during the early morning hours.

Since we do not directly measure Cl atom concentrations at our site, the midday peak of Cl-OOMs suggests that photochemical sources of Cl atoms beyond ClNO$_2$ photolysis may be active. Previous work has identified several daytime sources of reactive chlorine in polluted environments, including the photolysis of Cl$_2$, HOCl, ClONO$_2$, ICl, and BrCl, which can persist or increase under strong solar radiation later in the day (Peng et al., 2020; Ma et al., 2023). These sources may contribute to Cl atom production during midday, particularly in anthropogenically influenced regions like suburban Shanghai.

Additionally, we acknowledge the reviewer's suggestion that some Cl-containing compounds may be formed via reactions involving OH radicals with Cl-substituted VOCs or other intermediates, rather than directly from Cl atom-initiated reaction. While our current dataset does not allow us to fully resolve these mechanistic pathways, we now include a discussion in the revised manuscript addressing this alternative formation route and the potential contribution of both Cl- and OH-initiated processes to the observed Cl-OOMs.

We have revised our manuscript accordingly, which (Lines 468-477) reads,

"Similar to previous reports in northern Europe and Beijing (Breton et al., 2018; Priestley et al., 2018), Cl-OOMs increased with elevated solar radiation and their peaks appeared at around 12:00 p.m. (local time) hinting that the formation of Cl-OOMs is connected with photochemistry. This suggests that while ClNO$_2$ photolysis is a significant early morning source of Cl atoms, other sources such as the photolysis of Cl$_2$, ClONO$_2$, HCl, ICl, and BrCl by sunlight can also contribute to Cl atom concentrations later in the day (Peng et al., 2020). Besides, some Cl-OOMs could be formed through secondary reactions involving OH radicals with Cl-substituted VOCs

or intermediates, rather than direct Cl atom-initiated reactions. While distinguishing these pathways is beyond the scope of this study, the formation of Cl-OOMs is likely influenced by both Cl- and OH-initiated mechanisms under ambient conditions, especially in the presence of $NO_X$."

**References**

Breton, M. L., Hallquist, Å. M., Pathak, R. K., Simpson, D., Wang, Y., Johansson, J., Zheng, J., Yang, Y., Shang, D., Wang, H., Liu, Q., Chan, C., Wang, T., Bannan, T. J., Priestley, M., Percival, C. J., Shallcross, D. E., Lu, K., Guo, S., Hu, M., and Hallquist, M.: Chlorine oxidation of VOCs at a semi-rural site in Beijing: significant chlorine liberation from ClNO2 and subsequent gas- and particle-phase Cl–VOC production, Atmos Chem Phys, 18, 13013–13030, https://doi.org/10.5194/acp-18-13013-2018, 2018.

Bianchi, F., Kurtén, T., Riva, M., Mohr, C., Rissanen, M. P., Roldin, P., Berndt, T., Crounse, J. D., Wennberg, P. O., Mentel, T. F., Wildt, J., Junninen, H., Jokinen, T., Kulmala, M., Worsnop, D. R., Thornton, J. A., Donahue, N., Kjaergaard, H. G., and Ehn, M.: Highly Oxygenated Organic Molecules (HOM) from Gas-Phase Autoxidation Involving Peroxy Radicals: A Key Contributor to Atmospheric Aerosol, Chem Rev, 119, 3472–3509, https://doi.org/10.1021/acs.chemrev.8b00395, 2019.

Ehn, M., Thornton, J. A., Kleist, E., Sipilä, M., Junninen, H., Pullinen, I., Springer, M., Rubach, F., Tillmann, R., Lee, B., Lopez-Hilfiker, F., Andres, S., Acir, I.-H., Rissanen, M., Jokinen, T., Schobesberger, S., Kangasluoma, J., Kontkanen, J., Nieminen, T., Kurtén, T., Nielsen, L. B., Jørgensen, S., Kjaergaard, H. G., Canagaratna, M., Maso, M. D., Berndt, T., Petäjä, T., Wahner, A., Kerminen, V.-M., Kulmala, M., Worsnop, D. R., Wildt, J., and Mentel, T. F.: A large source of low-volatility secondary organic aerosol, Nature, 506, 476–479, https://doi.org/10.1038/nature13032, 2014.

Peng, X., Wang, W., Xia, M., Chen, H., Ravishankara, A. R., Li, Q., Saiz-Lopez, A., Liu, P., Zhang, F., Zhang, C., Xue, L., Wang, X., George, C., Wang, J., Mu, Y., Chen, J., and Wang, T.: An unexpected large continental source of reactive bromine and chlorine with significant impact on wintertime air quality, Natl. Sci. Rev., 8, nwaa304, https://doi.org/10.1093/nsr/nwaa304, 2020.

Priestley, M., Breton, M. le, Bannan, T. J., Worrall, S. D., Bacak, A., Smedley, A. R. D., Reyes-Villegas, E., Mehra, A., Allan, J., Webb, A. R., Shallcross, D. E., Coe, H., and Percival, C. J.: Observations of organic and inorganic chlorinated compounds and their contribution to chlorine radical concentrations in an urban environment in northern Europe during the wintertime, Atmos Chem Phys, 18, 13481–13493, https://doi.org/10.5194/acp-18-13481-2018, 2018.

---

## Author Response (AR2)

**RE: A point-to-point response to the reviewer's comments**

Journal: *Atmospheric Chemistry and Physics*

Manuscript No.: egusphere-2025-607

Title: "Formation of Chlorinated Organic Compounds from Cl Atom-Initiated Reactions of Aromatics and Their Detection in Suburban Shanghai"

Author(s): Chuang Li, Lei Yao, Yuwei Wang, Mingliang Fang, Xiaojia Chen, Lihong Wang, Yueyang Li, Gan Yang, Lin Wang

Dear editor,

We appreciate reviewers' thorough evaluation and valuable comments, which have been taken into account when improving our manuscript. In the following response, we have addressed Reviewer #2's comments in detail. Key updates include the addition of time series plots for a broader range of Cl-OOMs, as well as modifications to specific reaction mechanisms as suggested by Reviewer #2.

The reviewers' comments are repeated in *italics* while the responses are in normal fonts.

We are looking forward to your decision at your earliest convenience.

Sincerely,

Lei Yao

Lin Wang

**Reviewer #2:**

*Q0. I believe the authors have responded adequately to the initial comments from the first two reviewers. I also appreciate the authors' decision to conduct additional flow tube experiments using CO as an OH radical scavenger, which I believe is an effective way to isolate Cl and OH initiated chemistry within the system. However, looking through the authors' new Figure R1 and Scheme R2, I have some lingering questions about the underlying chemistry that I would like to have answered before recommending publication.*

**Response:** We appreciate *Reviewer #2's* positive feedback.

***Comments:***

*Q1: Can the authors show an alternate version of Figure R1 (either just for review or for the SI) that contains a larger number of the Cl-OOMs? I'm curious whether a larger subset of the Cl-OOMs identified in the paper have behavior similar to that of $C_8H_{11}ClO_2$ and don't show dependence towards OH reaction or more similar to that of $C_8H_{11}ClO_4$, which did show dependence towards OH reaction.*

**Response:** We appreciate the reviewer's helpful suggestion. In response, we have added an updated Figure R1 to incorporate a broader range of Cl-OOMs (i.e., $C_8H_{11}ClO_{2-7}$ and $C_8H_{13}ClO_{5-8}$).

As shown in the updated Figure R1, most Cl-OOMs (specifically $C_8H_{11}ClO_{3-7}$ and $C_8H_{13}ClO_{5-8}$) exhibit a significant decrease in signal intensity upon the addition of CO, indicating that their formation is influenced by the OH reaction. In contrast, $C_8H_{11}ClO_2$—a first-generation product derived directly from Cl-initiated oxidation—remains largely unaffected by OH suppression. This distinction highlights the differential roles of Cl and OH radicals in the formation of these species, with the majority of the expanded Cl-OOMs showing behavior more similar to that of $C_8H_{11}ClO_4$, which did exhibit dependence towards OH reaction.

[Figure]

**Figure R1.** Time-resolved signal intensities of Cl-OOMs products (panel A: $C_8H_{11}ClO_{2-7}$; panel B: $C_8H_{13}ClO_{5-8}$) from the Cl + m-xylene reaction in the absence of NOx. In the legend, the Cl-OOMs enclosed by solid lines are measured by I-CIMS, while those by dashed lines are measured by nitrate-CIMS.

*Q2: The Cl-OOM shown in Figure R1 that does not decrease in intensity following CO introduction to the flow tube, $C_8H_{11}ClO_2$, is shown in Scheme R2 as forming from an alkyl radical undergoing reaction with $HO_2$ radical. Given the presence of $O_2$ in the system, I don't believe this reaction should be likely. Can the authors posit another formation mechanism or provide rate estimations showing the feasibility of this reaction?*

**Response:** Thank you for raising this important point. We agree that the direct reaction between an alkyl radical and $HO_2$ is unlikely to be a dominant pathway. To resolve this, we propose another potential formation pathway for $C_8H_{11}ClO_2$, as illustrated in the revised Scheme R2. This updated mechanism draws inspiration from Bhattacharyya et al.'s work on the OH-initiated oxidation of m-xylene and suggests that $C_8H_{11}ClO_2$ may

form through the isomerization of the $C_8H_{10}ClO_2$ radical, followed by bond cleavage facilitated by electron rearrangements (Bhattacharyya et al., 2023). However, this formation pathway remains speculative, as we are currently unable to quantify its reaction rate due to the lack of available data. Consequently, we have chosen not to include this possible mechanism in the main text or Supplementary Information of the manuscript, to avoid overinterpretation.

Additionally, we performed smoothing on the real-time signal of $C_8H_{11}ClO_2$ (Figure R1(a)), and it can be observed that after CO addition, its signal shows an upward trend. Although this increase is not significant, it still indicates that the elevation of $HO_2$ promotes the formation of $C_8H_{11}ClO_2$.

**Scheme R2.** Proposed reaction pathways of first-generation products from Cl-initiated reactions of m-xylene. Blue and black formulae denote radicals and stable products, respectively.

*Q3: The Cl-OOM shown in Figure R1 that decreases in intensity following CO introduction to the flow tube, $C_8H_{11}ClO_4$, is shown in Scheme R2 as forming from reaction between an $RO_2$ radical and an $HO_2$ radical. The addition of CO should likely increase the concentration of $HO_2$ in the system, as CO effectively cycles OH into $HO_2$ radicals. This would presumably increase the amount of $C_8H_{11}ClO_4$ formed, as reaction*

*with $RO_2$ would be less likely, but the opposite effect is observed. Can the authors provide an explanation or alternate formation mechanism to rationalize the observed decrease in $C_8H_{11}ClO_4$ concentration following CO introduction?*

**Response:** We appreciate the reviewer's insightful observation, which highlights a key nuance in the system's chemistry. Indeed, the addition of CO is expected to convert OH radicals to $HO_2$, potentially enhancing the $Cl-RO_2 + HO_2$ pathway and thereby increasing Cl-OOM formation in principle. However, the observed decrease in the $C_8H_{11}ClO_4$ signal upon CO introduction (as shown in Figure R2) indicates that its production is influenced by OH-dependent pathways, overriding any potential benefits from elevated $HO_2$ levels.

To rationalize this, we propose that $C_8H_{11}ClO_4$ can form via multiple mechanisms. One pathway involves the reaction of the $C_8H_{10}ClO_4$ radical with $HO_2$, directly yielding $C_8H_{11}ClO_4$. An alternative and likely route is a secondary Cl-addition process: $C_8H_{10}O_4$ undergoes Cl addition to produce $C_8H_{11}ClO_4$. The introduction of CO suppresses OH availability, thereby reducing the formation of $C_8H_{10}O_4$ and limiting subsequent chlorination steps. This results in an overall decrease in $C_8H_{11}ClO_4$ concentration, despite the $HO_2$ increase.

This interpretation is supported by the experimental data in Figure R2, where both the $C_8H_{10}O_4$ (blue line) and $C_8H_{11}ClO_4$ (red line) signals rise sharply after UVA activation (indicating photolysis initiation) but decline following CO introduction. However, it should be noted that the decline in the $C_8H_{11}ClO_4$ signal is less pronounced than that of $C_8H_{10}O_4$; although we are currently unable to quantitatively describe these two species, their relative reduction extents suggest that $C_8H_{11}ClO_4$ has additional sources (namely, the reaction of the $C_8H_{10}ClO_4$ radical with $HO_2$).

[Figure]

**Figure R2.** Normalized signal intensities of $C_8H_{10}O_4$ (blue line) and $C_8H_{11}ClO_4$ (red line) as a function of reaction time in the flow tube experiment.

**Reference**

Bhattacharyya, N., Modi, M., Jahn, L. G., and Ruiz, L. H.: Different chlorine and hydroxyl radical environments impact m -xylene oxidation products, Environ. Sci.: Atmos., https://doi.org/10.1039/d3ea00024a, 2023.